# Understanding Generalization from Embedding Dimension and Distributional Convergence

**Junjie Yu** [* 1 2]   **Zhuoli Ouyang** [* 3]   **Haotian Deng** [* 1]   **Chen Wei** [1]   **Wenxiao Ma** [1]   **Jianyu Zhang** [1]   **Zihan Deng** [1]   **Quanying Liu** [1 2 4]

## Abstract

Deep neural networks often generalize well despite heavy over-parameterization, challenging classical parameter-based analyses. We study generalization from a representation-centric perspective and analyze how the geometry of learned embeddings is associated with generalization performance for a fixed trained model. We derive a post-hoc generalization bound that relates the gap between population risk and held-out empirical risk to two factors: (i) the intrinsic dimension of the embedding, which determines the convergence rate of the empirical embedding distribution to its population counterpart in Wasserstein distance, and (ii) the sensitivity of the downstream mapping from embeddings to predictions, characterized by Lipschitz constants. Together, these provide a post-hoc explanation of generalization for trained models. At the final embedding layer, architectural sensitivity disappears and the bound is dominated by embedding dimension, explaining its strong empirical correlation with generalization performance. Experiments across architectures and datasets validate the theory and demonstrate the utility of embedding-based diagnostics.

## 1. Introduction

Deep networks continue to grow in parameter count and training data, with empirical scaling laws (Kaplan et al., 2020) confirming that scale is a central driver of generalization. Yet this very trend exposes the inadequacy of classical generalization theory: bounds based on VC dimension (Vapnik et al., 1994; Sontag et al., 1998), Rademacher complexity (Bartlett & Mendelson, 2002), or norm (Bartlett et al., 2017; Neyshabur et al., 2015) become vacuous at modern scales and provide only worst-case guarantees over large hypothesis classes, failing to explain the generalization behavior of any individual trained model. What is needed instead is a measure that can rank concrete trained models and explain why certain training strategies generalize better, providing actionable guidance for training design.

These limitations motivate a representation-centric view of generalization, where the object of analysis is the geometry of embeddings induced by a trained model rather than its parameterization. Unlike parameter spaces, which grow unboundedly with model scale, embedding spaces remain compact and can be further standardized via PCA, making comparisons between models far less sensitive to differences in model size. Besides, representations capture the combined influence of data, optimization, and architecture, making them a richer and more holistic object of analysis than parameters alone.

Within this view, *intrinsic dimension* is a particularly useful geometric property of embedding. Although modern models produce high-dimensional embeddings, empirical studies show that these embeddings often concentrate near much lower-dimensional structures, and lower intrinsic dimension has been repeatedly associated with better generalization across architectures and training regimes (Ansuini et al., 2019; Pope et al., 2021). Yet several basic questions remain open: does intrinsic dimension vary meaningfully across layers, and if so, which layer is most diagnostic? Does dimension estimated on training data agree with that on test data, and what does any discrepancy reveal? And is intrinsic dimension alone sufficient to characterize generalization, or does it capture only part of the picture?

To address these questions, we derive a post-hoc generalization bound that makes the role of embedding dimension explicit. Unlike classical generalization bounds that provide uniform worst-case guarantees over entire hypothesis classes, our bound is instance-specific: it controls the gap between population risk and held-out empirical risk for a fixed,

[1]Department of Biomedical Engineering, Southern University of Science and Technology, Shenzhen, China [2]Omni-Intelligence, Shenzhen, China [3]Department of Electronic and Electrical Engineering, Southern University of Science and Technology, Shenzhen, China [4]Shenzhen Loop Area Institute , Shenzhen, China. Correspondence to: Quanying Liu <liuqy@sustech.edu.cn>.

*Proceedings of the 43rd International Conference on Machine Learning*, Seoul, South Korea. PMLR 306, 2026. Copyright 2026 by the author(s).

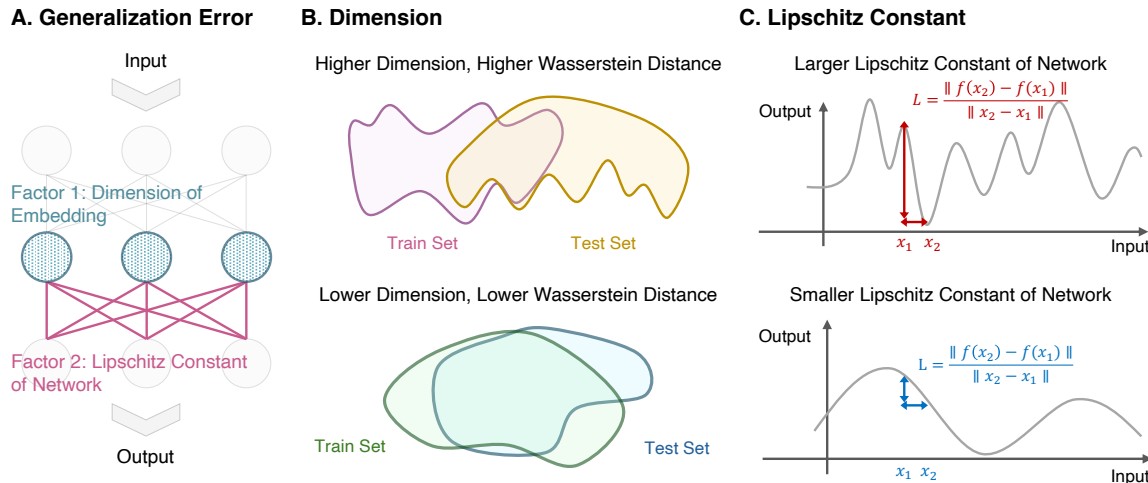

*Figure 1.* **Embedding Dimension and Network Lipschitz Constant Jointly Control the Post-hoc Generalization Gap. (A)** For a fixed trained model, the gap between population risk and held-out empirical risk is controlled jointly by the intrinsic dimension of the embedding distribution and the Lipschitz constant of the downstream network. **(B)** Lower intrinsic dimension leads to faster convergence of the empirical embedding distribution to its population counterpart in Wasserstein distance. **(C)** Smaller Lipschitz constants reduce the amplification of embedding perturbations into prediction or loss variations.

already-trained model. The central insight is that intrinsic dimension governs the rate at which the empirical embedding distribution converges to its population counterpart in Wasserstein distance. Lower intrinsic dimension implies lower sample complexity for approximating the population embedding distribution, enabling more reliable estimation of population risk from finite held-out samples.

Building on sharp convergence rate of Wasserstein distance (Weed & Bach, 2019), we show that for a fixed trained model $F$, each layer $k$ with intrinsic dimension $d_k$, Lipschitz constant $\bar{L}_k$, and distribution-dependent constants $C_k, D_k$, the population risk $R(F)$ and held-out empirical risk $\hat{R}_n(F)$ satisfy, with high probability over an independent held-out sample of size $n$,

$$R(F) \lesssim \hat{R}_n(F) + \bar{L}_k \left( C_k \, n^{-1/(d_k+\epsilon)} + D_k \sqrt{\frac{1}{n} \log \frac{L}{\delta}} \right),$$

where we display only the dominant terms. The term $n^{-1/(d_k+\epsilon)}$ characterizes the Wasserstein convergence rate of the held-out empirical embedding distribution to its population counterpart, while $\bar{L}_k$ quantifies how strongly downstream layers amplify embedding perturbations before they affect the loss (Figure 1). Crucially, this bound reveals that generalization improves when the embedding space has lower intrinsic dimension and when downstream layers have smaller Lipschitz constant, providing a concrete, model-specific criterion that classical worst-case bounds fail to capture.

The bound further clarifies which representations are most

informative for post-hoc generalization assessment. As $k$ approaches the output layer, $\bar{L}_k$ typically decreases, reaching exactly 1 at the final layer, thereby amplifying the influence of intrinsic dimension $d_k$ on the bound. This provides theoretical support for the widely observed empirical finding that final-layer intrinsic dimension is strongly associated with generalization performance.

Our contributions are threefold:

1. We derive a high-probability post-hoc generalization bound for fixed trained models, relating the gap between population risk and held-out empirical risk to the intrinsic dimension of embedding distributions.
2. We further show that, for the final layer, the bound reduces to a form primarily governed by the intrinsic dimension of the output representation.
3. We validate these theoretical predictions across large-scale vision and language models, demonstrating strong agreement between the bound and empirical observations.

## 2. Related Works

**Classical Generalization Bounds.** Theoretical analyses of generalization have traditionally focused on controlling hypothesis class capacity in parameter space. VC-dimension and Rademacher-complexity bounds (Vapnik & Chervonenkis, 2015; Bartlett & Mendelson, 2002) provide uniform worst-case guarantees, while margin- and norm-based refinements (Bartlett et al., 2017; Neyshabur et al., 2015; 2017) and PAC-Bayesian approaches (Arora et al., 2018; Hellström et al., 2025; Lotfi et al., 2022) tighten these estimates

via weight norms, or prior-posterior divergences. However, such bounds scale with parameter-space complexity and target uniform guarantees over model families, making them poorly suited to explaining the behavior of a particular trained model. Recent work has begun exploring learned representations as a more tractable alternative (Achille & Soatto, 2018; Papyan et al., 2020; Puli et al., 2021). Yet how embedding geometry formally connects to generalization remains largely unexplored.

**Representation-Based Approaches.** Recent research has increasingly focused on the impact of embedding geometry on generalization. Key approaches analyze properties such as consistency and separability of representations (Davies & Bouldin, 2009; Dyballa et al., 2024; Belcher et al., 2020). These geometric metrics offer improved interpretability and are less directly tied to model scale than parameter-based measures. However, they typically require labeled data, limiting their applicability in settings such as pretraining or self-supervised learning, where label information may be unavailable or only indirectly defined.

**Intrinsic Dimension of Representations.** A recent and promising direction in representation-based analysis is the study of *intrinsic dimension*, which quantifies the geometric complexity of embeddings. This approach aligns with the growing evidence that learned representations often concentrate near low-dimensional structures: lower intrinsic dimension can be interpreted as stronger geometric compression and has been empirically linked to improved generalization (Ansuini et al., 2019; Pope et al., 2021). While intrinsic dimension provides a label-free measure of representational complexity, the theoretical mechanisms connecting it to generalization remain largely unexplored. Our work addresses this gap by showing how intrinsic dimension controls the convergence of empirical embedding distributions to their population counterparts and how this error propagates through the downstream network to affect risk.

# 3. Preliminaries

This section introduces the key concepts, definitions, and assumptions that connect representation geometry to post-training generalization. Unless otherwise stated, empirical quantities in the theoretical analysis are computed on an independent held-out sample, and the trained predictor is treated as fixed.

## 3.1. Measures and Wasserstein distance

**Definition 3.1** (Empirical measure). Let $\mu$ be a probability distribution on a metric space $(X, d)$. Given $n$ i.i.d. samples

$\{x_i\}_{i=1}^{n} \sim \mu$, the empirical distribution is

$$\hat{\mu}_n = \frac{1}{n} \sum_{i=1}^{n} \delta_{x_i}.$$

**Definition 3.2** (Wasserstein distance). Let $(X, d)$ be a Polish metric space, and let $\alpha, \beta$ be probability measures on $X$ with finite first moments. The Wasserstein distance between $\alpha$ and $\beta$ is defined as

$$\mathcal{W}_1(\alpha, \beta) = \inf_{\gamma \in \Gamma(\alpha,\beta)} \int_{X \times X} d(x, y) \, d\gamma(x, y),$$

where $\Gamma(\alpha, \beta)$ denotes the set of probability measures on $X \times X$ with marginals $\alpha$ and $\beta$.

In our setting, $X = \mathcal{Z}_k$ is the embedding space equipped with the Euclidean ground metric $d(z, z') = \|z - z'\|_2$. Thus, throughout the paper, $\mathcal{W}_1$ denotes the Wasserstein-1 distance induced by the $\ell_2$ ground metric. The subscript 1 refers to the Wasserstein order, not to the $\ell_1$ norm. It quantifies how well the empirical embedding distribution $\hat{\tilde{P}}_{k,n}^{Z}$ approximates its population counterpart $\tilde{P}_{k}^{Z}$.

## 3.2. Network Decomposition and Embeddings

**Definition 3.3** (Network decomposition). At an intermediate layer $k$, the network is decomposed into an *encoder* $F_{\leq k} : \mathcal{X} \to \mathcal{Z}_k$ mapping an input $x \in \mathcal{X}$ to an embedding $z \in \mathcal{Z}_k$, and a *tail map* $F_k : \mathcal{Z}_k \to \mathbb{R}^C$ producing the final prediction. The overall predictor can thus be written as

$$F(x) = F_k\big(F_{\leq k}(x)\big).$$

**Definition 3.4** (Empirical and population embedding distributions). Given $n$ i.i.d. samples $\{x_i\}_{i=1}^{n} \sim P_X$, the *empirical embedding distribution* at layer $k$ is defined as

$$\hat{\tilde{P}}_{k,n}^{Z} = \frac{1}{n} \sum_{i=1}^{n} \delta_{F_{\leq k}(x_i)}.$$

The *population embedding distribution* $\tilde{P}_{k}^{Z}$ is the pushforward of $P_X$ under the embedding map,

$$\tilde{P}_{k}^{Z} = \mathbb{E}_{x \sim P_X}\big[\delta_{F_{\leq k}(x)}\big].$$

**Remark.** In the theoretical analysis, the predictor $F$ is first trained and then treated as fixed. The empirical embedding distribution $\hat{\tilde{P}}_{k,n}^{Z}$ is constructed from an independent held-out sample. In experiments, since $\tilde{P}_{k}^{Z}$ is not observable, we use embeddings from an independent test set as its empirical proxy. We do not use training-set embeddings for these quantities, because the embedding map $F_{\leq k}$ is itself learned from the training data, using the same data again would require additional arguments to account for this dependence.

## 3.3. Lipschitz continuity

**Definition 3.5** (Lipschitz continuity). A function $f : (X, d_X) \to (Y, d_Y)$ is $L$-Lipschitz if

$$d_Y\big(f(x), f(x')\big) \leq L\, d_X(x, x') \quad \text{for all } x, x' \in X.$$

The Lipschitz constant measures the sensitivity of the output to perturbations in the input and will be used to control how discrepancies in the embedding distribution propagate to the loss. Unless otherwise stated, all Lipschitz constants throughout the paper are defined with respect to the Euclidean $\ell_2$ metric.

## 3.4. Geometric Complexity and Wasserstein Convergence

To characterize the convergence of empirical measures in Wasserstein distance, we adopt the geometric framework of Weed & Bach (2019). The central insight of this framework is that convergence rate of Wasserstein distance is governed by the intrinsic dimension of the underlying distribution, formalized through the upper Wasserstein dimension.

**Definition 3.6** (Covering numbers and measure covering dimension). Let $(X, d)$ be a metric space and $S \subseteq X$. The $\varepsilon$–covering number of $S$ is

$$\mathcal{N}_\varepsilon(S) := \min \left\{ N : S \subseteq \bigcup_{i=1}^{N} B_i, \ \mathrm{diam}(B_i) \leq \varepsilon \right\}.$$

For a probability measure $\mu$ on $X$, the $(\varepsilon, \tau)$–covering number is

$$\mathcal{N}_\varepsilon(\mu, \tau) := \inf\{\mathcal{N}_\varepsilon(S) : \mu(S) \geq 1 - \tau\},$$

and the associated $(\varepsilon, \tau)$–dimension is

$$d_\varepsilon(\mu, \tau) := \frac{\log \mathcal{N}_\varepsilon(\mu, \tau)}{-\log \varepsilon}.$$

**Remark.** The quantity $d_\varepsilon(\mu, \tau)$ measures the effective geometric complexity of the bulk of the distribution at scale $\varepsilon$, while allowing a $\tau$–fraction of the probability mass to be ignored.

**Definition 3.7** (Upper Wasserstein dimension). For a probability measure $\mu$ on $(X, d)$ and $p \geq 1$, the *upper Wasserstein dimension* is defined as

$$d_p^*(\mu) := \inf \left\{ s > 2p : \limsup_{\varepsilon \to 0} d_\varepsilon\left(\mu, \ \varepsilon^{sp/(s-2p)}\right) \leq s \right\}.$$

**Remark.** The tolerance parameter $\tau = \varepsilon^{sp/(s-2p)}$ allows a vanishing fraction of the probability mass to be excluded as the scale $\varepsilon \to 0$, thereby preventing pathological, high-complexity regions of negligible mass from dominating the

dimension estimate. As shown in Weed & Bach (2019), the resulting quantity $d_p^*(\mu)$ precisely characterizes the minimax convergence rates of empirical measures in Wasserstein distance.

**Theorem 3.8** (Wasserstein convergence governed by intrinsic dimension). *Let $\hat{\mu}_n$ be the empirical measure of $n$ i.i.d. samples from a probability measure $\mu$ supported on a compact metric space. For any $p \in [1, \infty)$ and any $\varepsilon > 0$, setting $s = d_p^*(\mu) + \varepsilon$ yields*

$$\mathbb{E}\big[W_p(\mu, \hat{\mu}_n)\big] \leq C_{\varepsilon, p}\, n^{-1/s}.$$

*Since $\varepsilon$ may be chosen arbitrarily small, the convergence rate can be made arbitrarily close to $n^{-1/d_p^*(\mu)}$.*

**Remark.** Lower intrinsic dimension implies faster convergence of empirical distribution to underlying distribution. In the context of representation learning, estimating an intrinsic dimension proxy from embeddings therefore predicts how efficiently finite samples recover the population representation geometry.

## 3.5. Risk and Bayes Predictor

**Definition 3.9** (Population and empirical risk). For a fixed predictor $F : \mathcal{X} \to \mathbb{R}^C$ and loss function $\ell$, the *population risk* is

$$R(F) := \mathbb{E}_{(x,y) \sim P_{X,Y}}[\ell(F(x), y)],$$

and the *empirical risk* on $n$ i.i.d. samples $\{(x_i, y_i)\}_{i=1}^n$ is

$$\hat{R}_n(F) := \frac{1}{n} \sum_{i=1}^{n} \ell(F(x_i), y_i).$$

The quantity of interest in this work is the *generalization gap* $R(F) - \hat{R}_n(F)$ for a fixed, trained model $F$.

**Remark.** In experiments, the empirical risk is computed on the validation set, while the population risk is approximated using an independent test set.

**Definition 3.10** (Bayes predictor). Given a population distribution $P_{X,Y}$ over inputs and outputs, the *Bayes predictor* is defined as the conditional risk minimizer

$$F^*(x) := \arg\min_f \ \mathbb{E}\big[\ell(f(X), Y) \mid X = x\big].$$

It represents the population-optimal prediction achievable given access to the full data.

In the context of a fixed network and an intermediate layer $k$, we denote by $F_k^*$ the Bayes predictor associated with the embedding $Z_k = F_{\leq k}(X)$, corresponding to the population-optimal tail mapping from $\mathcal{Z}_k$ to the output space.

**Remark.** The Bayes predictor is a theoretical reference and is not assumed to be computable or learned. It is introduced

to separate approximation error due to information loss in the representation from estimation error due to finite samples. By replacing the discrete label $Y$ with the continuous target $F_k^*(Z_k)$, the loss becomes smooth with respect to the embedding distribution, enabling a distributional analysis. The resulting approximation error is explicitly accounted for in the generalization bounds.

### 3.6. Standing Assumptions

We now state the regularity assumptions required for the analysis, together with their roles in the proofs.

**Assumption 3.11** (Measurability of embeddings). For each layer $k$, the embedding map $F_{\leq k} : \mathcal{X} \to \mathcal{Z}_k$ is measurable, ensuring that the pushforward distribution $\tilde{P}_k^Z$ is well defined.

**Assumption 3.12** (Bounded support). Each embedding distribution $\tilde{P}_k^Z$ has bounded $\ell_2$-diameter:

$$D_k := \sup_{z,z' \in \mathrm{supp}(\tilde{P}_k^Z)} \|z - z'\|_2 < \infty.$$

The bounded diameter $D_k$ is used in Proposition A.9 (Appendix A.4.4) to control the effect of a single-sample replacement when applying McDiarmid's inequality to the Wasserstein term $W_1(\tilde{P}_k^Z, \hat{P}_{k,n}^Z)$.

**Assumption 3.13** (Local Lipschitz continuity of tail and Bayes maps). For each layer $k$, the network tail map $F_k$ and the Bayes predictor $F_k^*$ are assumed to be Lipschitz on the relevant embedding support, or on a convex neighborhood $U_k \supseteq \mathrm{supp}(\tilde{P}_k^Z)$. When the maps are differentiable on such a neighborhood, we use the bounds

$$L_k(F) := \sup_{z \in U_k} \|\nabla F_k(z)\|_{2 \to 2},$$

$$L_k(F^*) := \sup_{z \in U_k} \|\nabla F_k^*(z)\|_{2 \to 2}.$$

Here $\|\cdot\|_{2 \to 2}$ denotes the induced 2-norm (spectral norm) of the Jacobian. These constants control how perturbations in the embedding space propagate through the corresponding maps.

**Assumption 3.14** (Smooth loss with bounded gradient). The loss $\ell : \mathbb{R}^C \times \mathbb{R}^C \to \mathbb{R}$ is continuously differentiable in both arguments. There exist constants $M_F, M_{F^*} < \infty$ such that

$$\|\nabla_u \ell(u,v)\|_2 \leq M_F, \qquad \|\nabla_v \ell(u,v)\|_2 \leq M_{F^*}.$$

This assumption holds for squared error on any bounded relevant output range. Non-smooth losses (e.g., hinge loss) can be accommodated via standard smoothing arguments. These gradient bounds ensure that perturbations in the embeddings propagate in a controlled manner through the loss, making the subsequent analysis of embedding-induced errors feasible.

**Assumption 3.15** (Bounded Bayes surrogate noise). For each layer $k = 0, \ldots, L$, there exists a deterministic constant $B_u < \infty$ such that

$$\|Y - F_k^*(Z_k)\|_2 \leq B_u \quad \text{almost surely.}$$

Equivalently, the random variables $u_i^{(k)} := \|y_i - F_k^*(z_{k,i})\|_2$ are supported on $[0, B_u]$. This assumption is naturally satisfied in standard bounded-output settings, such as one-hot labels and probabilistic predictions.

## 4. Main Theoretical Results

### 4.1. Dimension-dependent post-hoc generalization bound

To formalize the role of low-dimensional representations in generalization, we derive a bound on the held-out-to-population gap $R(F) - \hat{R}_n(F)$ for a fixed trained model $F$, where $n$ denotes the held-out sample size. The bound explicitly captures how the intrinsic dimension of intermediate embeddings and the sensitivity of downstream mappings jointly control this discrepancy.

**Theorem 4.1** (Dimension-dependent post-hoc generalization bound). *Assume Assumptions 3.11–3.15, and fix a confidence level $\delta \in (0,1)$. For each layer $k$, let $d_k := d_1^*(\tilde{P}_k^Z)$ denote the upper Wasserstein dimension of the embedding distribution. Then for any $\epsilon > 0$, there exists a constant $C_k > 0$ such that, for all sufficiently large $n$,*

$$\mathbb{E}\left[\mathcal{W}_1(\tilde{P}_k^Z, \hat{\tilde{P}}_{k,n}^Z)\right] \leq C_k \, n^{-1/(d_k + \epsilon)}.$$

*Under these conditions, for any fixed predictor $F$, with probability at least $1 - \delta$,*

$$
\begin{aligned}
R(F) \leq &\, \hat{R}_n(F) \\
&+ \min_{0 \leq k \leq L} \left\{ \bar{L}_k \left( C_k \, n^{-1/(d_k + \epsilon)} + D_k \sqrt{\tfrac{1}{2n} \log \tfrac{2(L+1)}{\delta}} \right) \right. \\
&\left. + M_{F^*} \left( 2\,\mathbb{E}\|Y - F_k^*(Z_k)\|_2 + B_u \sqrt{\tfrac{1}{2n} \log \tfrac{2(L+1)}{\delta}} \right) \right\},
\end{aligned}
$$
(1)

*where $D_k$ is the $\ell_2$-diameter of the embedding support and*

$$\bar{L}_k := L_k(F)\, M_F + L_k(F^*)\, M_{F^*}.$$

*Here $L_k(F)$ denotes the Lipschitz constant of the network tail map from layer $k$ to the output, and $L_k(F^*)$ denotes the Lipschitz constant of the layer-wise Bayes predictor. $M_F$ and $M_{F^*}$ are the gradient bounds of the loss with respect to $F$ and $F^*$, as defined in Assumption 3.14. Together, these constants quantify how perturbations in the embeddings propagate through the network and the loss, enabling the analysis of embedding-induced generalization error.*

**Remarks.**

- **Dimension-controlled statistical error.** The dominant term $n^{-1/(d_k+\epsilon)}$ arises from the convergence rate of the empirical embedding distribution to its population counterpart in Wasserstein distance. Lower intrinsic dimension implies faster distributional convergence and thus smaller generalization gap for a fixed sample size.
- **Sensitivity amplification.** The factor $\bar{L}_k$ quantifies how discrepancies in the embedding distribution are amplified by downstream mappings and the loss. Even low-dimensional representations may generalize poorly if the predictor is highly sensitive to small embedding perturbations, highlighting the joint role of dimension and Lipschitz stability.
- **Bayes approximation (irreducible) error.** The term involving $\mathbb{E}\|Y - F_k^*(Z_k)\|_2$ captures the irreducible approximation error incurred when replacing discrete labels with the Bayes predictor.
- **Layer-wise tradeoff.** Different layers induce different balances between intrinsic dimension and sensitivity. Early layers may exhibit higher dimension but lower sensitivity, while later layers are typically more compressed but potentially more sensitive. Minimizing over $k$ selects the representation that provides the tightest control of the generalization gap.

### 4.2. Final-layer simplification

At the final layer, the representation coincides with the model output, $Z_L = F(X)$. In this case, the downstream mapping from embeddings to predictions is the identity map, and hence introduces no additional architectural amplification.

**Corollary 4.2** (Final-layer bound). *For the final embedding $Z_L$, the tail mapping is the identity and therefore $L_L(F) = 1$. With probability at least $1 - \delta$ over the independent held-out sample, conditional on the training sample,*

$$
\begin{aligned}
R(F) \;\le\; & \hat{R}_n(F) + \big(M_F + L_L(F^*)\,M_{F^*}\big) \\
& \cdot \Big(C_L\, n^{-1/(d_L+\epsilon)} + D_L\sqrt{\tfrac{1}{2n}\log\tfrac{2(L+1)}{\delta}}\Big) \\
& + M_{F^*}\Big(2\,\mathbb{E}\|Y - F_L^*(Z_L)\|_2 \\
& + B_u\sqrt{\tfrac{1}{2n}\log\tfrac{2(L+1)}{\delta}}\Big). \quad (2)
\end{aligned}
$$

**Remark.** At the final layer, architectural sensitivity disappears from the bound, leaving a dependence only on: (i) the intrinsic dimension $d_L$, (ii) the embedding diameter $D_L$, (iii) loss-dependent smoothness constants $(M_F, M_{F^*})$, and (iv) the Bayes smoothness and irreducible error terms. This simplification explains why final-layer intrinsic dimension often serves as a strong empirical predictor of generalization: performance is governed primarily by representation geom-

etry and data-dependent smoothness rather than network architecture.

**Summary.** Together, Theorem 4.1 and Corollary 4.2 formalize the value of low-dimensional representations: lower intrinsic dimension accelerates the convergence of the empirical embedding distribution to its population counterpart, directly tightening the held-out-to-population gap. This effect is modulated by the Lipschitz sensitivity of downstream mappings, and is most transparently isolated at the final layer. The complete proof is provided in Appendix A.

## 5. Experiments and Results

We empirically evaluate the mechanisms suggested by the theory in Section 4. Each experiment is designed to test a specific component of the post-hoc generalization bound: (i) the dimension–controlled Wasserstein convergence rate (Section 5.1), (ii) the final-layer simplification that removes architectural sensitivity (Section 5.2 and 5.3), and (iii) the joint role of embedding dimension and downstream Lipschitz sensitivity at intermediate layers (Section 5.4).

### 5.1. Validation of Wasserstein Convergence Scaling

Theorem 3.8 predicts that the convergence rate of empirical to population distributions in Wasserstein distance is governed by the intrinsic dimension of the underlying distribution. We first ask whether this scaling law holds for the complex, data-dependent embeddings produced by neural networks.

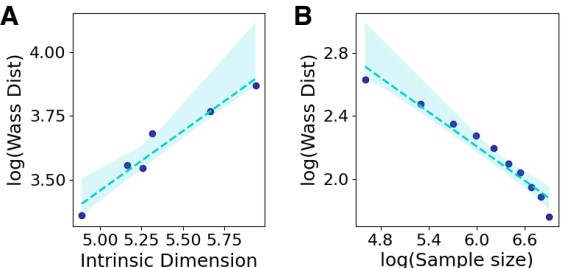

*Figure 2.* **Scaling of Wasserstein Convergence in Neural Network Embeddings.** **(A)** With fixed sample size, log(Wasserstein distance) increases approximately linearly with embedding dimension. **(B)** With fixed embedding dimension, log(Wasserstein distance) decreases approximately linearly with log(sample size).

We train a five-layer MLP autoencoder on MNIST and analyze how the Wasserstein distance between empirical and population embedding distributions depends on both intrinsic dimension and sample size. We examine two complementary perspectives. First, we fix the sample size $n$ and study how the Wasserstein distance varies with the intrinsic dimension of the embedding. Second, we fix the intrinsic dimension and evaluate how the Wasserstein distance scales

with $n$ according to the predicted $n^{-1/(d+\epsilon)}$ rate. Sample sizes are varied from $n = 100$ to $n = 1500$. For each configuration, intrinsic dimension is estimated using the MLE estimator of Levina & Bickel (2004).

The results exhibit two consistent patterns. For fixed $n$, the Wasserstein distance increases approximately exponentially with intrinsic dimension, as predicted by the dimension-dependent rate (Figure 2A). Conversely, for fixed $d$, the Wasserstein distance decreases approximately as a power law in $n$, closely matching the theoretical $n^{-1/(d+\epsilon)}$ scaling (Figure 2B). These findings confirm that intrinsic dimension accurately governs Wasserstein convergence even for learned neural network embeddings. Additional experimental details are provided in Appendix B.1.

### 5.2. Analysis of Dimension, Wasserstein Distance and Generalization in Convolutional Networks

Corollary 4.2 predicts that when analyzing the final layer of a network, architectural sensitivity vanishes and the generalization gap exposes the dimension-dependent Wasserstein term more directly, up to Bayes smoothness, loss, diameter, and surrogate-noise constants. We empirically test this prediction across architectures and datasets.

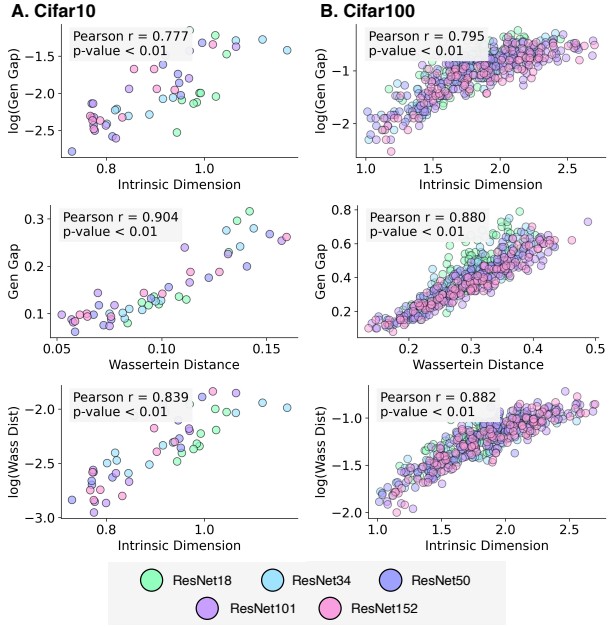

*Figure 3.* **Relationship Between Final-Layer Embedding Dimension, Wasserstein Distance and Generalization Error.** We evaluate CIFAR-10 (**A**) and CIFAR-100 (**B**) and observe a significant correlation between final-layer embedding dimension, Wasserstein distance and generalization error.

We evaluate ResNet-18, 34, 50, 101, and 152 on CIFAR-10 and CIFAR-100. For each trained model, we extract the final-layer embeddings and estimate their intrinsic dimen-

sion. We also compute the empirical Wasserstein distance between validation and test embedding distributions, which approximates the distributional discrepancy appearing in the final-layer bound.

To obtain finer-grained statistical resolution, we perform the analysis at the class level. Each ResNet model thus yields 10 data points on CIFAR-10 and 100 data points on CIFAR-100, allowing us to assess the relationship between embedding geometry and generalization gap across a broad range of conditions.

Figure 3 shows a clear positive relationship between final-layer intrinsic dimension, empirical Wasserstein distance, and the generalization gap. Because the network-tail sensitivity becomes the identity at the final layer, this result supports the corollary's mechanism and shows that representation geometry provides a strong post-hoc signal for comparing generalization behavior across architectures. Additional analyses and robustness checks are provided in Appendix B.2, Appendix C and Appendix F.

### 5.3. Analysis of Dimension, Wasserstein Distance, and Generalization in Large Models

We further examine whether the representation-geometric relationships suggested by Corollary 4.2 persist in large pretrained models and large-scale benchmarks. We analyze pretrained vision models on ImageNet-1K and pretrained language models on MNLI (Williams et al., 2018). Since the original training data and validation losses are unavailable, we do not directly estimate the held-out-to-population gap. Instead, we use benchmark accuracy as the performance measure and compute Wasserstein distance between embeddings of two disjoint subsets from the evaluation dataset, which serves as a proxy for the finite-sample stability of the embedding distribution. Architectural details are deferred to Appendix D.

As shown in Figure 4, the qualitative pattern observed in controlled experiments also appears in this large-model setting. For both vision and language models, lower final-layer intrinsic dimension is associated with smaller subset-to-subset Wasserstein distance, indicating more similar empirical embedding distributions across different subsets. Both quantities also correlate with performance: models with lower intrinsic dimension and smaller Wasserstein distance generally achieve higher ImageNet-1K or MNLI accuracy.

These results show that the relationship between embedding dimension and Wasserstein distributional convergence holds at the scale of large pretrained models, and that both quantities are significantly associated with generalization performance. This suggests that final-layer embedding geometry provides a practical post-training diagnostic for model comparison, even when training details are unavailable.

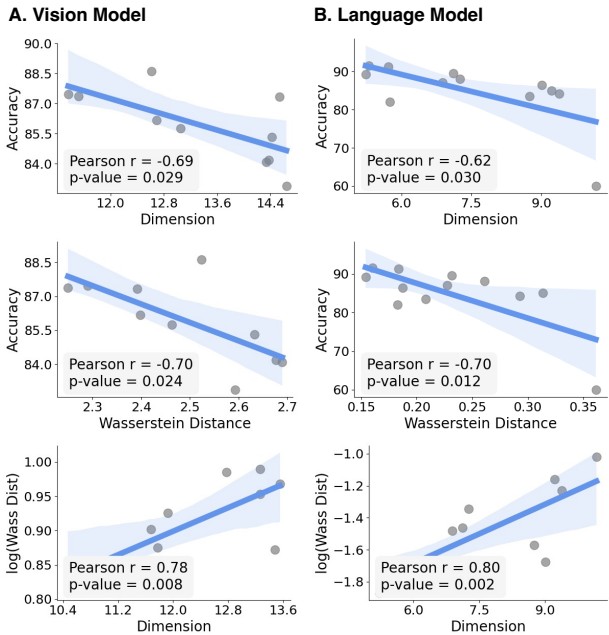

Figure 4. **Relationship between embedding geometry and generalization performance in large models.** (**A**) Vision models. (**B**) Language models. Across both modalities, lower final-layer intrinsic dimension is associated with smaller Wasserstein distance and stronger performance.

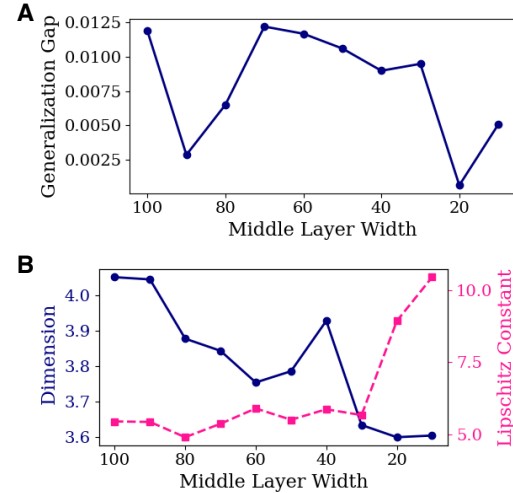

Figure 5. **Effect of Network Width on Embedding Dimension and Generalization.** (**A**) Reducing the width of the third layer does not lead to a consistent decrease in generalization error. (**B**) As the layer width decreases, the embedding dimension gradually decreases, but the network's Lipschitz constant increases, offsetting potential gains.

## 5.4. Interventions on Network Width

Theorem 4.1 predicts that at intermediate layers, generalization depends jointly on embedding intrinsic dimension and the Lipschitz sensitivity of the downstream mapping. To empirically test this interaction, we perform controlled architectural interventions that selectively modify one factor while tracking the other.

Directly computing the Lipschitz constant of deep networks is intractable in general. We therefore consider fully connected MLPs with ReLU activations, for which the product of spectral norms of the weight matrices provides a computable upper bound on the network's Lipschitz constant (Bartlett et al., 2017). We use this quantity as a sensitivity proxy and vary the width of a single intermediate layer to study its effect on both embedding geometry and sensitivity.

Specifically, we train a six-layer MLP on CIFAR-10 and vary the width of the third layer from 100 to 10. Figure 5 shows that while narrowing the layer consistently reduces the intrinsic dimension of the embedding, it also increases the network's Lipschitz constant, particularly at small widths. As a result, the generalization gap does not improve monotonically.

These results provide causal evidence for the tradeoff predicted by our theory: generalization is jointly governed by representation geometry and the sensitivity of

the downstream mapping. While narrowing the network enforces lower-dimensional embeddings, it can simultaneously increase the Lipschitz constant of the learned mapping, thereby offsetting potential gains in generalization. This explains why architectural compression alone does not guarantee improved generalization and highlights the necessity of considering both embedding dimensionality and network stability in overparameterized models. Additional details are provided in Appendix B.3.

## 6. Discussion and Conclusion

Our work provides a theoretical explanation for why the intrinsic dimension of learned embeddings is informative for generalization. Lower-dimensional embedding distributions converge faster from empirical to population distributions, so finite held-out samples provide a more reliable estimate of population risk. This makes validation-based checkpoint selection more faithful to population-level model selection. The same perspective also helps explain why low-dimensional embeddings often yield better generalization performance: when the embedding distribution has low intrinsic dimension, finite independent samples can more effectively cover its relevant geometry, making the learned embedding-to-output mapping more likely to transfer to unseen data. Beyond dimension, our bound shows that this distributional approximation error is further modulated by the Lipschitz constant $\bar{L}_k$, which controls how strongly embedding-space discrepancies are amplified into the loss. Together, intrinsic dimension and Lipschitz regularity pro-

vide a post-hoc, representation-geometric account of the held-out-to-population gap.

Our framework may be also relevant for latent generative models such as VAEs, representation autoencoders, and latent diffusion models, where both representation compactness and downstream sensitivity are central (Bond-Taylor et al., 2021; Zheng et al., 2025; Rombach et al., 2022). In these settings, low-dimensional latent distributions can improve coverage and distributional estimation from finite samples, while the decoder or generative mapping must remain stable under latent perturbations that move beyond exact training examples. Excessive downstream sensitivity may amplify small off-support deviations into severe semantic or visual artifacts. From this perspective, our analysis suggests that favorable latent spaces should not only exhibit low intrinsic dimension, but also support sufficiently regular embedding-to-output mappings. We believe this provides a natural representation-geometric perspective on the tradeoff between latent compactness and generative robustness in modern latent generative models.

In Appendix E, we analyze layer-wise embeddings in ResNet-152 by computing correlations among intrinsic dimension, Wasserstein distance, and generalization performance. We find that intrinsic dimension and Wasserstein distance are already strongly correlated in early layers, while the correlations between dimension and generalization, as well as between Wasserstein distance and generalization, increase progressively with network depth. This supports our theoretical prediction that low-dimensional embeddings improve distributional estimation, and that this effect becomes more directly connected to generalization in later layers where the downstream mapping is shorter and less sensitive.

**Limitations.** A fundamental constraint of our framework stems from the dependency between the trained model and its training-set embeddings. Since model parameters are optimized on the training data, the resulting embeddings are statistically dependent on the model itself, violating the i.i.d. assumption underlying our distributional convergence analysis. This prevents us from using training-set embeddings or training-set empirical risk in our bound, and explains why our bound differs structurally from classical generalization bounds that relate training-set quantities to population risk. Moreover, because the analysis is performed post hoc on a fixed trained model, the resulting bound characterizes the geometry of the learned representation rather than the dynamics of the optimization process itself, and therefore does not directly capture the influence of the training algorithm, optimization trajectory, or implicit regularization effects. Addressing these dependencies, for instance via stability-based techniques (Bousquet & Elisseeff, 2002; Charles & Papailiopoulos, 2018), is a key step toward extending our

framework into a standard generalization bound in the conventional sense.

On the technical side, the constants in our bound, most notably the downstream Lipschitz constant $\bar{L}_k$, can be difficult to estimate reliably for modern architectures and may render the bound loose in practice. Developing scalable and sample-efficient estimators for $\bar{L}_k$ therefore remains an important direction for future work. Besides, under severe label noise, the Bayes surrogate noise terms can dominate the Wasserstein term, making the bound loose in a way that reflects irreducible label uncertainty.

**Conclusion.** This work shows that low-dimensional learned representations are useful because they make the data distribution in embedding space easier to approximate from finite samples. By connecting intrinsic dimension to convergence of empirical distribution, we provide a theoretical mechanism for why compact embeddings often correlate with better generalization. Our results suggest that late-layer intrinsic dimension can serve as a practical diagnostic for comparing trained models and understanding how representation geometry contributes to their performance.

## Acknowledgements

This work was supported by Brain Science and Brain-like Intelligence Technology - National Science and Technology Major Project (2021ZD0200500), the National Natural Science Foundation of China (62472206, 3254100307), National Key R&D Program of China (2025YFC3410000), Shenzhen Science and Technology Innovation Committee (RCYX20231211090405003, JCYJ20220818100213029), Guangdong Basic and Applied Basic Research Foundation (2026B1515020099), Guangdong S&T Program (Grant No. 2026B0101110003), Shanghai Municipal Special Program for Basic Research on General AI Foundation Models (2025SHZDZX026D05), GuangDong Basic and Applied Basic Research Foundation (2025A1515011645), GuangDong Basic and Applied Basic Research Foundation (2026A1515010121), Shenzhen Doctoral Startup Project (RCBS20231211090748082), and the open research fund of the Guangdong Provincial Key Laboratory of Mathematical and Neural Dynamical Systems, the Center for Computational Science and Engineering at Southern University of Science and Technology, Shenzhen Key Laboratory of Smart Healthcare Engineering.

## Impact Statement

This work aims to advance the understanding of generalization in deep learning by analyzing how embedding geometry and network sensitivity influence performance. Our findings primarily provide theoretical and practical insights for de-

signing and analyzing neural networks. We do not anticipate any direct negative societal consequences from this research. While advances in machine learning can have broad societal implications, the present study is foundational in nature, and any potential downstream applications will depend on how these insights are used in specific contexts.

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

# Overview of Supplementary Materials

This supplementary material provides detailed derivations, proofs, and experimental analyses that complement the main text. Its purpose is to give a complete and self-contained presentation of our theoretical results, as well as extensive experimental validation of the proposed framework. The content can be broadly organized into two main parts: theoretical analysis and experimental details.

**1. Theoretical Analysis (Sections A).** We derive a high-probability, dimension-dependent bound for neural networks, emphasizing the role of intermediate embedding geometry and network Lipschitz properties. Key steps include:

- **Notation and preliminaries (Sectio A.1):** Introduce all key symbols and collect standard results on optimal transport and Wasserstein distances.
- **Risk decomposition via Bayes surrogates (Section A.2):** Split the generalization gap into (A) approximation, (B) oracle statistical, and (C) empirical model terms, enabling Lipschitz-based analysis.
- **Bounding each component (Section A.4):** Control each term using deterministic Lipschitz arguments and concentration inequalities, producing high-probability bounds that scale with embedding dimension and sample size.
- **Recovering network effects (Section A.5):** Decompose the oracle Lipschitz constant to separate controllable network contributions from intrinsic distribution-dependent effects, yielding interpretable insights on architecture influence.

**2. Experimental Analyses (Sections B — F).** We provide comprehensive experimental validations to illustrate the theoretical insights and investigate practical aspects of embeddings and network design. The main experimental directions are:

- **Synthetic embedding experiments (Section B.1):** Study how Wasserstein distances depend on intrinsic dimension and sample size using MNIST autoencoders.
- **Cross-architecture analysis (Section B.2):** Analyze embedding distributions and class-wise generalization gaps across multiple ResNet architectures on CIFAR-10/100.
- **Causal width experiments (Section B.3):** Explore how hidden layer width affects embedding dimensionality, network Lipschitz constants, and generalization in MNIST MLPs.
- **Dimensionality estimation and hyperparameter sensitivity (Section C):** Assess the influence of hyperparameters and estimation methods on intrinsic dimension measurements.
- **Large-scale pretrained models (Section D):** Detail the pretrained vision and language models used in our evaluation, spanning diverse architectures, scales, and pretraining regimes.
- **Layer-wise correlation analysis (Section E):** Analyze how embedding dimensionality, Wasserstein distance, and generalization error correlate across layers of ResNet-152, showing stronger alignment in deeper layers.
- **Predictive power of intrinsic dimension and Wasserstein distance (Section F):** Assess whether intrinsic dimension and Wasserstein distance from smaller ResNets can predict generalization gaps on unseen larger architectures.

Together, these theoretical and empirical components provide a comprehensive understanding of the interplay between embedding geometry, network design, and generalization, complementing and extending the results presented in the main paper.

# A. Supplement of Theoretical Results

Before presenting the detailed proofs, we first summarize the key notation used throughout this appendix and the main paper. This notation table serves as a convenient reference to improve clarity and readability.

**Notation summary.** Key symbols used throughout the paper.

*Table 1.* Notation summary for key symbols in the paper.

| Symbol | Meaning |
|---|---|
| $\tilde{P}_k^Z$ | Population embedding distribution at layer $k$. |
| $\hat{\tilde{P}}_{k,n}^Z$ | Empirical embedding distribution from $n$ samples. |
| $\hat{\mu}_n$ | Empirical measure of $n$ i.i.d. samples. |

| Symbol | Meaning |
| --- | --- |
| $R(F)$ | Population risk of predictor $F$. |
| $\hat{R}_n(F)$ | Empirical risk on validation set. |
| $\text{gen}(F)$ | Generalization gap $R(F) - \hat{R}_n(F)$. |
| $F_{\leq k}$ | Encoder mapping input $x$ to embedding $z$ at layer $k$. |
| $F_k$ | Tail map from layer-$k$ embedding $z$ to output. |
| $F(x)$ | Overall predictor $F_k(F_{\leq k}(x))$. |
| $F_k^*$ | Bayes predictor from layer-$k$ embedding $z$ to output. |
| $d_k$ | Intrinsic dimension of $\tilde{P}_k^Z$. |
| $D_k$ | $\ell_2$-diameter of support of $\tilde{P}_k^Z$. |
| $\mathcal{W}_1(\cdot, \cdot)$ | 1-Wasserstein distance. |
| $L_k(F)$ | Lipschitz constant of network tail from layer $k$ to output. |
| $L_k(F^*)$ | Lipschitz constant of Bayes predictor from layer $k$ to output. |
| $M_F$ | Bound on loss gradient w.r.t network output. |
| $M_{F^*}$ | Bound on loss gradient w.r.t Bayes predictor output. |
| $B_u$ | Uniform upper bound on the Bayes surrogate noise $\|Y - F_k^*(Z_k)\|_2$. |
| $\ell$ | Loss function (e.g., squared loss). |
| $B_\ell$ | Uniform bound on loss values. |

**Roadmap of the appendix.** This appendix provides a complete derivation of the dimension-dependent post-hoc generalization bound stated in Theorem 4.1. The proof is organized into four main steps:

1. **Preliminaries (Subsection A.1):** We collect standard technical tools used throughout the proofs, including optimal transport results and Wasserstein bounds for Lipschitz functions.
2. **Risk decomposition via Bayes surrogates (Subsection A.2):** In classification settings, labels are discrete, so the observed loss is non-differentiable with respect to embeddings. We introduce layer-wise Bayes predictors as continuous surrogates, leading to a decomposition of the generalization error into three terms: (A) approximation gap, (B) oracle statistical gap, and (C) empirical model gap.
3. **Controlling the decomposed terms (Subsection A.4):** Each term is bounded explicitly. (A) and (C) are controlled by irreducible label noise, while (B) is controlled via the 1-Wasserstein distance between empirical and population embeddings combined with the oracle loss Lipschitz constant. Concentration inequalities yield high-probability bounds scaling with embedding dimension and sample size.
4. **Recovering network effects (Subsection A.5):** The oracle Lipschitz constant $L_k(g)$ is decomposed as

$$L_k(g) \leq L_k(F)\, M_F + L_k(F^*)\, M_{F^*},$$

separating controllable network-dependent and intrinsic Bayes predictor contributions. Substituting this into the previous bounds connects embedding geometry, statistical concentration, and network design.

Overall, these steps provide a clear, high-probability generalization bound that disentangles statistical, architectural, and label-noise contributions.

## A.1. Preliminaries: Useful Lemmas and Theorems

In this subsection we collect several standard results that will be used throughout the proofs. They are presented here to avoid interruptions in the main arguments later.

### A.1.1. EXISTENCE OF OPTIMAL TRANSPORT PLAN

**Theorem A.1** (Existence of Optimal Transport Plan (Villani et al., 2009))**.** *Let $(\mathcal{X}, \mu)$ and $(\mathcal{Y}, \nu)$ be Polish probability spaces, and let*

$$c : \mathcal{X} \times \mathcal{Y} \longrightarrow \mathbb{R} \cup \{+\infty\}$$

*be a lower semicontinuous cost function. Then there exists a coupling $\gamma^* \in \Pi(\mu, \nu)$ that minimizes the expected cost:*

$$\int_{\mathcal{X} \times \mathcal{Y}} c(x, y) \, d\gamma^*(x, y) = \inf_{\gamma \in \Pi(\mu, \nu)} \int_{\mathcal{X} \times \mathcal{Y}} c(x, y) \, d\gamma(x, y).$$

*In particular, for the 1-Wasserstein cost $c(z, z') = \|z - z'\|_2$ on a Polish space there exists an optimal coupling attaining $\mathcal{W}_1$.*

### A.1.2. WASSERSTEIN BOUND FOR LIPSCHITZ FUNCTIONS

**Lemma A.2** (Expectation difference controlled by $W_1$)**.** *Let $(\mathbb{R}^d, \|\cdot\|_2)$ be equipped with the $\ell_2$ metric, and let $\mu, \nu$ be probability measures with finite first moments. If $h : \mathbb{R}^d \to \mathbb{R}$ is $L_h$-Lipschitz with respect to $\ell_2$, i.e.,*

$$|h(z) - h(z')| \leq L_h \|z - z'\|_2 \quad \forall z, z',$$

*then*

$$\left| \int h \, d\mu - \int h \, d\nu \right| \leq L_h \, \mathcal{W}_1(\mu, \nu).$$

*Proof.* This follows directly from the Kantorovich–Rubinstein dual representation of $\mathcal{W}_1$ on Polish metric spaces.

By definition of $\mathcal{W}_1$ and for any coupling $\pi \in \Pi(\mu, \nu)$,

$$\int h \, d\mu - \int h \, d\nu = \iint \big( h(z) - h(z') \big) \, d\pi(z, z') \leq \iint L_h \|z - z'\|_2 \, d\pi(z, z').$$

Taking infimum over all couplings $\pi$ gives the claim. The absolute value follows by symmetry (swapping $\mu, \nu$). $\qquad\square$

### A.2. Risk Decomposition via Bayes Surrogates

**Motivation.** In classification, the label $Y$ is discrete, so the observed loss $\ell(F(X), Y)$ is not differentiable with respect to embeddings $Z_k$. This obstructs a direct Lipschitz-based analysis of the risk, which is central to our approach. To address this, we introduce at each layer $k$ the *Bayes predictor* $F_k^*(Z_k)$, a continuous surrogate for the discrete label. Replacing $Y$ with $F_k^*(Z_k)$ yields the *oracle loss*, which is differentiable in $Z_k$ and hence amenable to Lipschitz/Wasserstein analysis. The cost of this replacement is an additional error term capturing the mismatch between observed and oracle risks. This term corresponds to irreducible label randomness and will be explicitly controlled.

**Definition A.3** (Observed and oracle risks)**.** Let $\ell : \mathbb{R}^C \times \mathbb{R}^C \to \mathbb{R}$ be a measurable loss. At the input layer, the observed risks are

$$R_0^{\text{obs}} := \mathbb{E}_{(X,Y) \sim \mathcal{D}}[\ell(F(X), Y)], \qquad \hat{R}_{0,n}^{\text{obs}} := \frac{1}{n} \sum_{i=1}^n \ell(F(x_i), y_i).$$

At layer $k$, the oracle loss is defined as

$$g_k(z) := \ell(F_k(z), F_k^*(z)),$$

with population and empirical oracle risks

$$R_k^{\text{oracle}} := \mathbb{E}_{Z_k \sim \tilde{P}_k^Z}[g_k(Z_k)], \qquad \hat{R}_{k,n}^{\text{oracle}} := \frac{1}{n} \sum_{i=1}^n g_k(z_{k,i}).$$

The following identity is purely algebraic and holds deterministically for any fixed dataset and predictor.

**Proposition A.4** (Risk decomposition)**.** *For any predictor $F$ and any intermediate layer $k$,*

$$R_0^{\text{obs}} - \hat{R}_{0,n}^{\text{obs}} = (R_0^{\text{obs}} - R_k^{\text{oracle}}) + (R_k^{\text{oracle}} - \hat{R}_{k,n}^{\text{oracle}}) + (\hat{R}_{k,n}^{\text{oracle}} - \hat{R}_{0,n}^{\text{obs}}).$$

**Interpretation.** The decomposition separates the generalization gap into three terms:

- **Approximation gap:** $R_0^{\text{obs}} - R_k^{\text{oracle}}$ measures the loss of information when replacing discrete labels by the Bayes predictor at layer $k$.
- **Oracle statistical gap:** $R_k^{\text{oracle}} - \hat{R}_{k,n}^{\text{oracle}}$ is the population-to-sample deviation of the oracle loss, the term to be controlled via Lipschitz continuity and Wasserstein concentration.
- **Empirical model gap:** $\hat{R}_{k,n}^{\text{oracle}} - \hat{R}_{0,n}^{\text{obs}}$ quantifies how network predictions differ from Bayes-optimal predictions under the empirical distribution.

**Summary.** The observed generalization error is thus expressed as an oracle component (amenable to Lipschitz/Wasserstein analysis) plus two additional error terms that capture irreducible label noise and model approximation. This motivates analyzing the Lipschitz constant of the oracle loss $g_k(z)$, which we do next.

## A.3. Lipschitz Constant of the Layer-Wise Loss

Having introduced the oracle loss $g_k(z) = \ell(F_k(z), F_k^*(z))$, we now analyze its Lipschitz continuity with respect to the embedding $z$ under the Euclidean metric. This is possible because both arguments of $g_k$ are differentiable functions of $z$.

**Gradient and Lipschitz bound.**

**Lemma A.5.** *Suppose Assumptions 3.11–3.14 hold. Assume that $F_k$ and $F_k^*$ are differentiable on $U_k$, and that the loss $\ell$ has bounded Euclidean gradients:*

$$\|\partial_F \ell(u,v)\|_2 \leq M_F, \qquad \|\partial_{F^*} \ell(u,v)\|_2 \leq M_{F^*}$$

*for all relevant $(u,v)$. Then for all $z \in U_k$,*

$$\nabla g_k(z) = \nabla F_k(z)^\top \, \partial_F \ell(F_k(z), F_k^*(z)) + \nabla F_k^*(z)^\top \, \partial_{F^*} \ell(F_k(z), F_k^*(z)).$$

*Moreover,*

$$\|\nabla g_k(z)\|_2 \ \leq \ \|\nabla F_k(z)\|_{2\to 2} \, \|\partial_F \ell(F_k(z), F_k^*(z))\|_2 + \|\nabla F_k^*(z)\|_{2\to 2} \, \|\partial_{F^*} \ell(F_k(z), F_k^*(z))\|_2.$$

*Consequently, if the Jacobians satisfy*

$$\|\nabla F_k(z)\|_{2\to 2} \leq L_k(F), \qquad \|\nabla F_k^*(z)\|_{2\to 2} \leq L_k(F^*)$$

*for all $z \in U_k$, then the Euclidean Lipschitz constant of $g_k$ satisfies*

$$L_k(g) := \sup_{z \in U_k} \|\nabla g_k(z)\|_2 \ \leq \ L_k(F) \, M_F + L_k(F^*) \, M_{F^*}.$$

*Equivalently, for all $z, z' \in U_k$,*
$$|g_k(z) - g_k(z')| \leq L_k(g) \, \|z - z'\|_2.$$

*Proof.* The chain rule gives

$$\nabla g_k(z) = \nabla F_k(z)^\top \, \partial_F \ell(F_k(z), F_k^*(z)) + \nabla F_k^*(z)^\top \, \partial_{F^*} \ell(F_k(z), F_k^*(z)).$$

Taking Euclidean norms and applying the triangle inequality yields

$$\|\nabla g_k(z)\|_2 \leq \|\nabla F_k(z)^\top \partial_F \ell(F_k(z), F_k^*(z))\|_2 + \|\nabla F_k^*(z)^\top \partial_{F^*} \ell(F_k(z), F_k^*(z))\|_2.$$

Using the spectral norm bound

$$\|A^\top v\|_2 \leq \|A^\top\|_{2\to 2}\|v\|_2 = \|A\|_{2\to 2}\|v\|_2,$$

we obtain

$$\|\nabla g_k(z)\|_2 \leq \|\nabla F_k(z)\|_{2\to 2}\|\partial_F \ell(F_k(z), F_k^*(z))\|_2 + \|\nabla F_k^*(z)\|_{2\to 2}\|\partial_{F^*} \ell(F_k(z), F_k^*(z))\|_2.$$

Substituting the uniform bounds gives

$$\|\nabla g_k(z)\|_2 \leq L_k(F)M_F + L_k(F^*)M_{F^*}.$$

Taking the supremum over $z \in U_k$ yields

$$L_k(g) = \sup_{z \in U_k} \|\nabla g_k(z)\|_2 \leq L_k(F) M_F + L_k(F^*) M_{F^*}.$$

Finally, since $g_k$ is differentiable on $U_k$, the mean-value inequality implies

$$|g_k(z) - g_k(z')| \leq L_k(g)\|z - z'\|_2.$$

This proves the claim. $\qquad \square$

**Remark.** The bound cleanly separates two contributions: (i) the network-dependent Euclidean Lipschitz constant $L_k(F)$, which can be controlled by architecture or regularization, and (ii) the Euclidean Lipschitz constant $L_k(F^*)$, reflecting the inherent complexity of the oracle or Bayes predictor. Thus the oracle loss Lipschitz constant decomposes into a controllable and an uncontrollable component, which will play distinct roles in the final generalization bound.

### A.4. Controlling the Decomposed Terms

**Overview of the approach.** Proposition A.4 decomposes the generalization gap into three terms:

$$\underbrace{R_0^{\mathrm{obs}} - R_k^{\mathrm{oracle}}}_{(A) \text{ approximation gap}}, \quad \underbrace{R_k^{\mathrm{oracle}} - \hat{R}_{k,n}^{\mathrm{oracle}}}_{(B) \text{ oracle statistical gap}}, \quad \underbrace{\hat{R}_{k,n}^{\mathrm{oracle}} - \hat{R}_{0,n}^{\mathrm{obs}}}_{(C) \text{ empirical model gap}} .$$

We now control these terms separately:

- (A) measures the error incurred by replacing labels $Y$ with the Bayes surrogate $F_k^*(Z_k)$.
- (B) measures the statistical deviation between population and empirical distributions of embeddings, for the oracle loss.
- (C) measures the discrepancy between network predictions and Bayes-optimal predictions under the empirical distribution.

Each of (A), (B), (C) will be treated in turn.

#### A.4.1. BOUNDING THE APPROXIMATION GAP (A).

**Lemma A.6** (Control of approximation gap). *Assume the loss $\ell : \mathbb{R}^C \times \mathbb{R}^C \to \mathbb{R}$ is Lipschitz in its second argument with constant $M_{F^*}$ (Assumption 3.14). Then for any predictor $F$ and any layer $k$,*

$$\left| R_0^{\mathrm{obs}} - R_k^{\mathrm{oracle}} \right| \leq M_{F^*} \, \mathbb{E}_{Z \sim \tilde{P}_k^Z} \left[ \|Y - F_k^*(Z)\|_2 \right].$$

*Proof.* For any sample $(x, y)$ with embedding $z = F_{\leq k}(x)$,

$$\left| \ell(F(x), y) - \ell(F_k(z), F_k^*(z)) \right| \leq M_{F^*} \|y - F_k^*(z)\|_2,$$

by Lipschitz continuity of $\ell$ in the second argument. Taking expectation over $(X, Y) \sim \mathcal{D}$ yields the result. $\qquad \square$

#### A.4.2. BOUNDING THE ORACLE STATISTICAL GAP (B).

**Lemma A.7** (Oracle risk controlled by $W_1$). *For any predictor $F \in \mathcal{F}$ and any layer $k$,*

$$\left| R_k^{\mathrm{oracle}} - \hat{R}_{k,n}^{\mathrm{oracle}} \right| \leq L_k(g) \, \mathcal{W}_1\big(\tilde{P}_k^Z, \hat{\tilde{P}}_{k,n}^Z\big).$$

*where $L_k(g)$ is the Lipschitz constant of $g_k(z) = \ell(F_k(z), F_k^*(z))$ with respect to the $\ell_2$-metric, as given in Lemma A.5.*

*Proof.* By Kantorovich-Rubinstein duality, for any $L$-Lipschitz function $f$,

$$\left| \int f \, d\mu - \int f \, d\nu \right| \leq L \, W_1(\mu, \nu).$$

Applying this with $f = g_k$, $\mu = \tilde{P}_k^Z$, and $\nu = \hat{\tilde{P}}_{k,n}^Z$, and recalling that $g_k$ has Lipschitz constant $L_k(g)$, gives the desired bound. $\qquad \square$

A.4.3. BOUNDING THE EMPIRICAL MODEL GAP (C).

**Lemma A.8** (Control of empirical model gap). *Under the same assumptions as Lemma A.6, for any predictor $F$ and any layer $k$,*

$$\left| \hat{R}_{k,n}^{\text{oracle}} - \hat{R}_{0,n}^{\text{obs}} \right| \leq M_{F^*} \frac{1}{n} \sum_{i=1}^{n} \| y_i - F_k^*(z_{k,i}) \|_2.$$

*Proof.* For each validation sample $(x_i, y_i)$ with embedding $z_{k,i} = F_{\leq k}(x_i)$,

$$\left| \ell(F(x_i), y_i) - \ell(F_k(z_{k,i}), F_k^*(z_{k,i})) \right| \leq M_{F^*} \| y_i - F_k^*(z_{k,i}) \|_2.$$

Averaging over $i = 1, \dots, n$ yields the result. $\square$

A.4.4. CONCENTRATION OF $T_k := \mathcal{W}_1(\tilde{P}_k^Z, \hat{\tilde{P}}_{k,n}^Z)$ AND OF THE EMPIRICAL NOISE AVERAGE

**Motivation.** The deterministic decomposition in Proposition A.4 reduces the generalization gap to three terms. Among them, two depend explicitly on the randomness of the empirical sample:

- the Wasserstein distance $T_k = \mathcal{W}_1(\tilde{P}_k^Z, \hat{\tilde{P}}_{k,n}^Z)$, which controls the oracle statistical gap (B);
- the empirical noise average $\bar{u}^{(k)} = \frac{1}{n} \sum_{i=1}^{n} \| y_i - F_k^*(z_{k,i}) \|_2$, which appears in the empirical model gap (C).

To obtain a high-probability generalization bound, it is therefore crucial to quantify how much these quantities deviate from their expectations. We now prove two concentration inequalities: a bounded-difference bound (McDiarmid) for $T_k$, and a Hoeffding bound for $\bar{u}^{(k)}$.

**Proposition A.9** (Concentration of $T_k$ and $\bar{u}^{(k)}$). *Let $D_k := \sup_{z,z' \in \text{supp}(\tilde{P}_k^Z)} \| z - z' \|_2 < \infty$ be the $\ell_2$-diameter of the embedding support. Define $T_k = \mathcal{W}_1(\tilde{P}_k^Z, \hat{\tilde{P}}_{k,n}^Z)$, and $\bar{u}^{(k)} = \frac{1}{n} \sum_{i=1}^{n} u_i^{(k)}$ with $u_i^{(k)} = \| y_i - F_k^*(z_{k,i}) \|_2$. By Assumption 3.15, $u_i^{(k)} \in [0, B_u]$ for all $i$. Then for any $\delta \in (0,1)$, with probability at least $1 - \frac{\delta}{L+1}$,*

$$T_k \leq \mathbb{E}[T_k] + D_k \sqrt{\frac{1}{2n} \log \frac{2(L+1)}{\delta}}, \tag{3}$$

$$\bar{u}^{(k)} \leq \mathbb{E}[u^{(k)}] + B_u \sqrt{\frac{1}{2n} \log \frac{2(L+1)}{\delta}}. \tag{4}$$

*Proof.* **Step 1: Bounded-difference inequality for $T_k$.** We use the Kantorovich-Rubinstein dual representation of $W_1$:

$$\mathcal{W}_1(\mu, \nu) = \sup_{\substack{f:\mathbb{R}^d \to \mathbb{R} \\ \text{Lip}(f) \leq 1}} \left\{ \int f \, d\mu - \int f \, d\nu \right\},$$

with Lipschitz constant measured in the $\ell_2$-norm. Let the empirical measure be $\hat{\tilde{P}}_{k,n}^Z = \frac{1}{n} \sum_{i=1}^{n} \delta_{z_{k,i}}$. Consider two samples $S = (z_{k,1}, \dots, z_{k,n})$ and $S^{(j)}$ that differ only in the $j$-th element. Denote $T_k(S) = \mathcal{W}_1(\tilde{P}_k^Z, \hat{\tilde{P}}_{k,n}^Z(S))$. Then

$$|T_k(S) - T_k(S^{(j)})| \leq \tfrac{1}{n} \| z_{k,j} - z_{k,j}' \|_2 \leq \tfrac{D_k}{n}.$$

Thus $T_k$ satisfies a bounded-difference property with sensitivity $D_k/n$. Applying McDiarmid's inequality gives, for any $t > 0$,

$$\mathbb{P}\big(T_k - \mathbb{E}[T_k] \geq t\big) \leq \exp\Big( -\frac{2nt^2}{D_k^2} \Big).$$

Choosing $t = D_k \sqrt{\frac{1}{2n} \log \frac{2(L+1)}{\delta}}$ yields (3).

**Step 2: Hoeffding bound for $\bar{u}^{(k)}$.** Each $u_i^{(k)} \in [0, B_u]$ by Assumption 3.15. By Hoeffding's inequality, for any $t > 0$,

$$\mathbb{P}\big(\bar{u}^{(k)} - \mathbb{E}[\bar{u}^{(k)}] \geq t\big) \leq \exp\Big( -\frac{2nt^2}{B_u^2} \Big).$$

Choosing $t = B_u \sqrt{\frac{1}{2n} \log \frac{2(L+1)}{\delta}}$ yields (4). A union bound over the two deviations gives the stated probability.

This completes the proof. $\square$

**Discussion.** This result ensures that both the statistical fluctuation of the embedding distribution (through $T_k$) and the empirical noise magnitude (through $\bar{u}^{(k)}$) remain close to their expectations with high probability. These concentration bounds are the key probabilistic ingredients needed to convert the deterministic decomposition of the generalization gap into a high-probability generalization bound.

### A.4.5. COMBINED DETERMINISTIC AND HIGH-PROBABILITY BOUND

**Motivation.** We now combine the pieces developed above. Recall that the observed generalization gap

$$R_0^{\text{obs}} - \hat{R}_{0,n}^{\text{obs}}$$

was decomposed into three terms (Proposition A.4). We provided deterministic bounds for each term (Lemmas A.6–A.8), and then concentration inequalities for the random quantities $T_k$ and $\bar{u}^{(k)}$ (Proposition A.9). Here we integrate these ingredients into a single high-probability generalization bound. All probabilistic statements below are conditional on the training sample, the only randomness is the independent held-out sample.

**Proposition A.10** (High-probability control of the generalization gap). *Assume Assumptions 3.11–3.15. Suppose that for each layer $k$ there exist constants $C_k > 0$, arbitrarily small $\epsilon > 0$ and $d_k > 0$ such that $\mathbb{E}[T_k] \leq C_k n^{-1/(d_k+\epsilon)}$ for all sufficiently large $n$, where the expectation is over the independent held-out sample conditional on the training sample. Fix confidence $\delta \in (0,1)$. Then with probability at least $1 - \delta$ over the independent held-out sample, simultaneously for all layers $k = 0, \ldots, L$ and for any fixed trained predictor independent of this held-out sample,*

$$R_0^{\text{obs}} - \hat{R}_{0,n}^{\text{obs}} \leq L_k(g)\left(C_k n^{-1/(d_k+\epsilon)} + D_k\sqrt{\tfrac{1}{2n}\log\tfrac{2(L+1)}{\delta}}\right)$$
$$+ M_{F^*}\left(2\,\mathbb{E}\|Y - F_k^*(Z)\|_2 + B_u\sqrt{\tfrac{1}{2n}\log\tfrac{2(L+1)}{\delta}}\right). \tag{5}$$

*In terms of rates and up to multiplicative constants, the bound can be summarized as*

$$\boxed{R_0^{\text{obs}} - \hat{R}_{0,n}^{\text{obs}} \;\lesssim\; L_k(g)\,n^{-1/(d_k+\epsilon)} \;+\; M_{F^*}\,\mathbb{E}\|Y - F_k^*(Z)\|_2 \;+\; \sqrt{\tfrac{\log(2(L+1)/\delta)}{n}}\left(L_k(g)D_k + B_u M_{F^*}\right)}$$

*Proof.* **Step 1: Decomposition.** By Proposition A.4,

$$R_0^{\text{obs}} - \hat{R}_{0,n}^{\text{obs}} = (A) + (B) + (C).$$

**Step 2: Deterministic bounds.** From Lemmas A.6 – A.8,

$$R_0^{\text{obs}} - \hat{R}_{0,n}^{\text{obs}} \leq M_{F^*}\,\mathbb{E}\|Y - F_k^*(Z)\|_2 \;+\; L_k(g)\,T_k \;+\; M_{F^*}\,\bar{u}^{(k)}.$$

**Step 3: Concentration.** Applying Proposition A.9 to each layer $k = 0, \ldots, L$ and taking a union bound over layers, the following inequalities hold simultaneously for all layers with probability at least $1 - \delta$:

$$T_k \leq \mathbb{E}[T_k] + D_k\sqrt{\tfrac{1}{2n}\log\tfrac{2(L+1)}{\delta}}, \qquad \bar{u}^{(k)} \leq \mathbb{E}[u^{(k)}] + B_u\sqrt{\tfrac{1}{2n}\log\tfrac{2(L+1)}{\delta}}.$$

Since $\mathbb{E}[u^{(k)}] = \mathbb{E}\|Y - F_k^*(Z)\|_2$, substituting yields

$$R_0^{\text{obs}} - \hat{R}_{0,n}^{\text{obs}} \leq L_k(g)\left(\mathbb{E}[T_k] + D_k\sqrt{\tfrac{1}{2n}\log\tfrac{2(L+1)}{\delta}}\right)$$
$$+ M_{F^*}\left(2\mathbb{E}\|Y - F_k^*(Z)\|_2 + B_u\sqrt{\tfrac{1}{2n}\log\tfrac{2(L+1)}{\delta}}\right).$$

Finally substitute $\mathbb{E}[T_k] \leq C_k n^{-1/(d_k+\epsilon)}$ to obtain (5). $\qquad\square$

**Discussion.** This bound highlights three components:

- The *statistical rate* $L_k(g)\,C_k n^{-1/(d_k+\epsilon)}$ combines embedding geometry (via $d_k$) and oracle loss sensitivity (via $L_k(g)$).
- The *noise/approximation terms* $M_{F^*}\,\mathbb{E}\|Y - F_k^*(Z)\|_2$ arise from replacing discrete labels by the Bayes predictor.
- The *concentration terms* scale as $O(\sqrt{\tfrac{\log(2(L+1)/\delta)}{n}})$, with constants depending on both distributional ($M_{F^*}$) and geometric ($D_k$) quantities.

Together, these yield an explicit and interpretable high-probability upper bound on the observed generalization gap.

### A.5. Recovering Network Effects via Lipschitz Constants

In the previous subsection, the oracle statistical gap (B) was controlled using the Lipschitz constant $L_k(g)$ of the oracle loss. We now expand it to expose how the bound depends both on the network architecture (controllable) and on the data distribution (intrinsic).

#### A.5.1. EXPANSION OF $L_k(g)$

From Lemma A.5,

$$L_k(g) := \sup_{z \in U_k} \|\nabla g_k(z)\|_2 \ \leq \ L_k(F) \, M_F + L_k(F^*) \, M_{F^*},$$

where:

- $L_k(F)$ is the Lipschitz constant of the tail sub-network from layer $k$ to the output;
- $L_k(F^*)$ is the Lipschitz constant of the Bayes predictor at layer $k$;
- $M_F$, $M_{F^*}$ are uniform derivative bounds of the loss with respect to its two arguments.

**Proof sketch.** By the chain rule,

$$\nabla g_k(z) = \nabla F_k(z)^\top \, \partial_F \ell(F_k(z), F_k^*(z)) + \nabla F_k^*(z)^\top \, \partial_{F^*} \ell(F_k(z), F_k^*(z)).$$

Applying the operator norm inequality and the uniform derivative bounds yields the stated inequality.

#### A.5.2. CONTROLLABLE VS. INTRINSIC CONTRIBUTIONS

This decomposition separates the two sources of sensitivity:

- **Network-dependent term:** $L_k(F) \, M_F$, which is determined by the architecture and training of the tail network. It can be reduced by explicit design choices (e.g., normalization layers, spectral norm constraints, Lipschitz regularization).
- **Distribution-dependent term:** $L_k(F^*) \, M_{F^*}$, which reflects the smoothness of the Bayes predictor relative to embeddings. This term is intrinsic to the data distribution and cannot be improved by network design.

#### A.5.3. IMPLICATION FOR THE POST-HOC GENERALIZATION BOUND

Substituting the decomposition of $L_k(g)$ into Proposition A.10 gives

$$R_0^{\text{obs}} - \hat{R}_{0,n}^{\text{obs}} \leq \left( L_k(F) \, M_F + L_k(F^*) \, M_{F^*} \right) \left( C_k n^{-1/(d_k+\epsilon)} + D_k \sqrt{\frac{1}{2n} \log \frac{2(L+1)}{\delta}} \right)$$
$$+ \underbrace{\left[ M_{F^*} \left( 2\mathbb{E}\|Y - F_k^*(Z)\|_2 + B_u \sqrt{\tfrac{1}{2n} \log \tfrac{2(L+1)}{\delta}} \right) \right]}_{\text{Bayes surrogate terms}}. \tag{6}$$

Thus the final bound reflects two complementary mechanisms:

1. *Embedding geometry:* the intrinsic dimension $d_k$ governs the statistical rate of Wasserstein convergence;
2. *Network design:* the Lipschitz constant $L_k(F)$ controls how embedding perturbations are amplified through the network;
3. *Bayes surrogate terms:* a residual contribution capturing the discrepancy between discrete labels and their Bayes predictor surrogate, including irreducible randomness.

## B. Details of Experiments

### B.1. Details of Section 5.1

We conducted an experiment on MNIST to study how the Wasserstein distance between empirical embedding distributions depends on (i) the intrinsic dimension of the embeddings and (ii) the number of samples used to estimate the distributions.

**Model and training.** We trained simple fully connected autoencoders with a symmetric architecture. The encoder flattened each $28 \times 28$ image and mapped it to 256 hidden units with ReLU activation, followed by a linear layer to a $d$-dimensional bottleneck. The decoder mirrored this with a linear layer back to 256 units, ReLU, and a final linear layer to 784 units. Training used mean squared error loss, the Adam optimizer with learning rate $10^{-3}$, batch size 128, and 30 epochs. Global randomness was controlled by setting a fixed seed for both PyTorch and NumPy.

**Data and embeddings.** All data were drawn from the MNIST dataset. For the analysis of intrinsic dimension, we trained autoencoders with bottleneck sizes $d \in \{16, 32, 64, 128, 256, 512\}$. For the analysis of sample size, we trained a single autoencoder with bottleneck dimension 64 and repeatedly drew two independent subsets of size $n \in \{100, 200, \ldots, 1000\}$ to evaluate how the Wasserstein distance scales with $n$. In all cases, embeddings from the training split were used as the empirical distribution, and embeddings from the test split were used as the population distribution. This experiment is intended as a diagnostic scaling study of Wasserstein behavior in learned embeddings, rather than a direct instantiation of the held-out generalization bound above.

**Intrinsic dimension estimation.** We estimated the intrinsic dimension using the maximum likelihood estimator of Levina and Bickel (Levina & Bickel, 2004), implemented in `skdim`.

**Wasserstein distance.** We measured discrepancies between embedding sets using an entropically regularized optimal transport cost (Sinkhorn distance). Uniform weights were assigned to all points, the ground cost was the Euclidean distance, and the regularization parameter was $\varepsilon = 10^{-2}$. Iterations terminated either after 200 steps or once the update magnitude fell below $10^{-6}$. The resulting cost was computed as the expectation of the ground cost under the transport plan.

## B.2. Details of Section 5.2

We conducted experiments on CIFAR-10 and CIFAR-100 to analyze how the final-layer embeddings relate to class-wise generalization gaps modified across ResNet architectures.

**Model and training.** We considered five ResNet architectures: ResNet-18, 34, 50, 101, and 152. Each model was initialized with ImageNet-pretrained weights from `torchvision.models` and evaluated on CIFAR datasets. The architecture of these ResNet models was modified by adjusting the final linear output layer. Specifically, the output of the model's convolutional layers is initially projected to a 128-dimensional space via a linear layer. This is then followed by a Sigmoid activation function, and finally, another projection layer yields the ultimate output. These nets are finetuning on Cifar-10 and Cifar-100 used the Adam optimizer with weight decay 0.001, base learning rate $10^{-4}$, and a cosine annealing schedule over 50 epochs. Batch size was 256, with random horizontal flip for augmentation. Multi-GPU training was enabled via `accelerate`. Models were saved after training and evaluated on the full test set.

**Embedding extraction.** For each trained model, we extracted the *last layer embeddings* for all samples in both validation and test splits. Embeddings were stored separately for each class to allow class-wise analysis. For the CIFAR-10 dataset, each class of embeddings in both the validation and test sets comprises 500 samples. In the case of the CIFAR-100 dataset, each class of embeddings in both the validation and test sets consists of 100 samples.

**Intrinsic dimension estimation.** We estimated the intrinsic dimension of these embeddings using the maximum likelihood estimator of Levina and Bickel (Levina & Bickel, 2004), as implemented in `skdim`. Estimates were computed independently for each class and averaged across samples, yielding 10 estimates per model on CIFAR-10 and 100 per model on CIFAR-100.

**Wasserstein distance.** For each class, we computed the Wasserstein distance between validation and test embeddings. This used entropic-regularized optimal transport (Sinkhorn distance) with Euclidean ground cost, uniform weights, and regularization parameter $\epsilon = 10^{-2}$. These distances quantify how far apart the validation and test embedding distributions are.

**Generalization gap.** For each class and model, validation and test losses were recorded to compute the class-wise generalization gap.

## B.3. Details of Section 5.4

We designed a experiment on MNIST to analyze how the width of a hidden layer influences intrinsic dimension of intermediate embeddings, Lipschitz properties of the network and generalization performance. The experiment uses a six-layer multilayer perceptron (MLP) with configurable hidden-layer widths and records both statistical and geometric properties of representations throughout training.

**Model and training.** The model is a fully connected network with architecture

$$784 \rightarrow h_1 \rightarrow h_2 \rightarrow h_3 \rightarrow h_4 \rightarrow h_5 \rightarrow 10,$$

where each hidden layer is followed by a ReLU activation. The default hidden width is 100 units for all layers. To study the effect of representation bottlenecks, we varied the width of the third hidden layer ($h_3$) over the list $\{100, 90, 80, 70, 60, 50, 40, 30, 20, 10\}$, while keeping all other layers fixed at 100. Training was performed with cross-entropy loss, the Adam optimizer (learning rate $10^{-3}$, weight decay 0), batch size 128, and for 10 epochs. We used both validation and test splits of MNIST, with additional evaluation on a fixed random subset of 2048 validation samples. All randomness was controlled by fixed seeds and deterministic settings in PyTorch to ensure reproducibility.

**Activation collection and intrinsic dimension.** To measure representation complexity across layers we registered forward hooks after each ReLU activation. During evaluation, the hooks collected activations for all inputs in the 2048-sample subset. For each layer's activation matrix $X$, we applied the maximum likelihood estimator (as implemented in `skdim`).

**Lipschitz estimation.** To characterize stability of the mapping from each hidden layer to the output, we computed the product of spectral norms of all subsequent linear layers. For a given suffix starting at layer $i$, the Lipschitz constant was approximated by

$$L_{i \rightarrow \text{end}} = \prod_{j=i+1}^{L} \sigma_{\max}(W_j),$$

where $W_j$ denotes the weight matrix of linear layer $j$ and $\sigma_{\max}$ is its top singular value. Singular values were computed using `torch.linalg.svdvals` in double precision. These suffix-wise Lipschitz estimates were recorded at initialization and after each epoch.

## C. Dimensionality Estimation and Hyperparameter Analysis

In this appendix, we investigate the effects of hyperparameter choices and the specific algorithm used on the estimation of embedding dimensionality. All experiments are conducted using subsets of the CIFAR datasets: 500 samples per class for CIFAR-10 and 100 samples per class for CIFAR-100.

### C.1. Hyperparameter Analysis

We first examine how the choice of the hyperparameter $K$ affects dimensionality estimates. Here, $K$ corresponds to the number of nearest neighbors used in the estimation procedure: larger $K$ values capture dimensionality over a broader range of the data, whereas smaller $K$ values reflect more local structure.

For CIFAR-10, we test $K = 100, 200, 300, 400, 500$, and for CIFAR-100, we test $K = 20, 40, 60, 80, 100$. Figures 6 and 7 show the results. We observe that as $K$ increases, the estimated dimensionality better correlates with the generalization error. This indicates that the global dimensionality of a class is more predictive of generalization performance than local dimensionality.

### C.2. Algorithm Comparison

Next, we compare different dimensionality estimation algorithms (TLE (Amsaleg et al., 2019) and MOM (Amsaleg et al., 2018)) while keeping the hyperparameter fixed ($K = 400$ for CIFAR-10, $K = 80$ for CIFAR-100).

Figures 8 and 9 present the results. Across both datasets, all algorithms yield estimated dimensionalities that remain significantly correlated with generalization error, suggesting that the observed relationship is robust to the choice of estimation method.

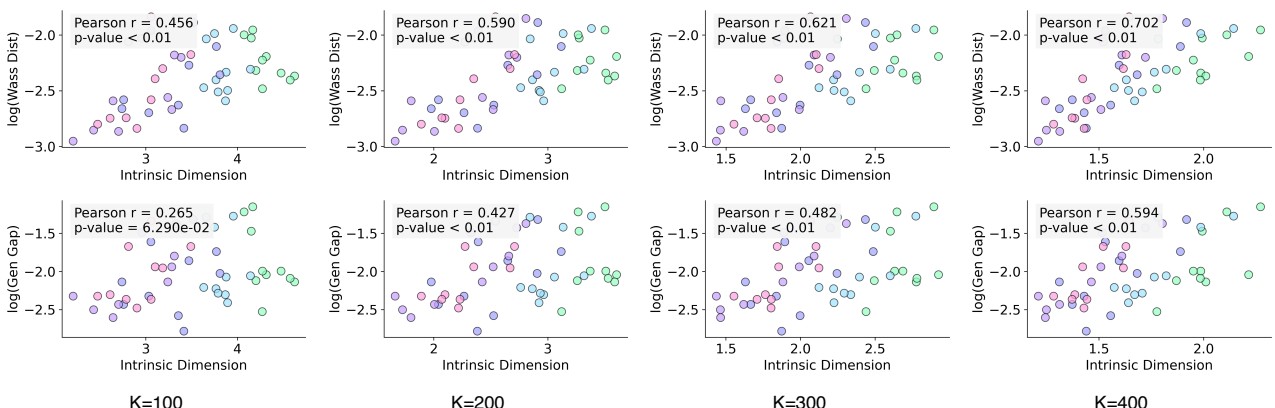

*Figure 6.* **Effect of hyperparameter $K$ on dimensionality estimation for CIFAR-10 embeddings.** Larger $K$ values capture broader data structure and lead to higher correlation with generalization error.

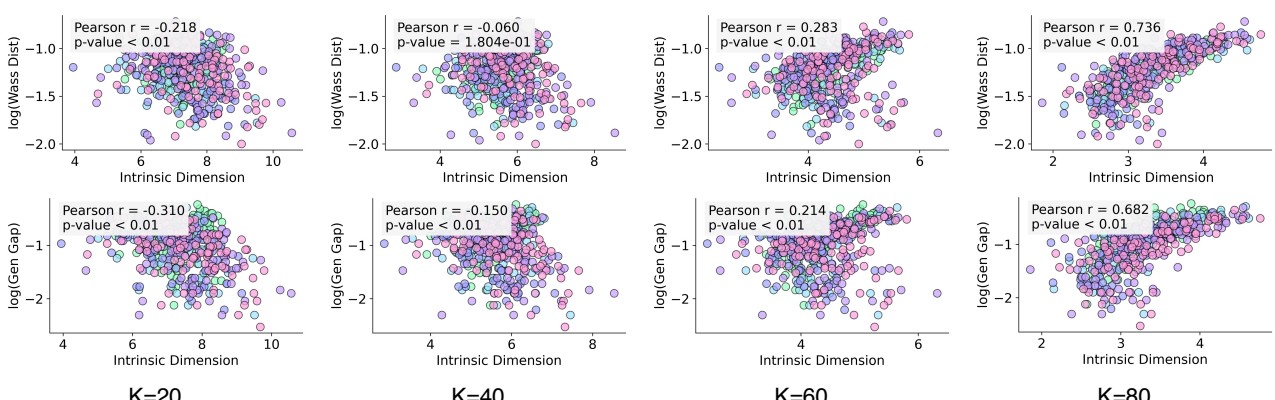

*Figure 7.* **Effect of hyperparameter $K$ on dimensionality estimation for CIFAR-100 embeddings.** Increasing $K$ improves the alignment between estimated dimensionality and generalization error, indicating that global structure is more informative.

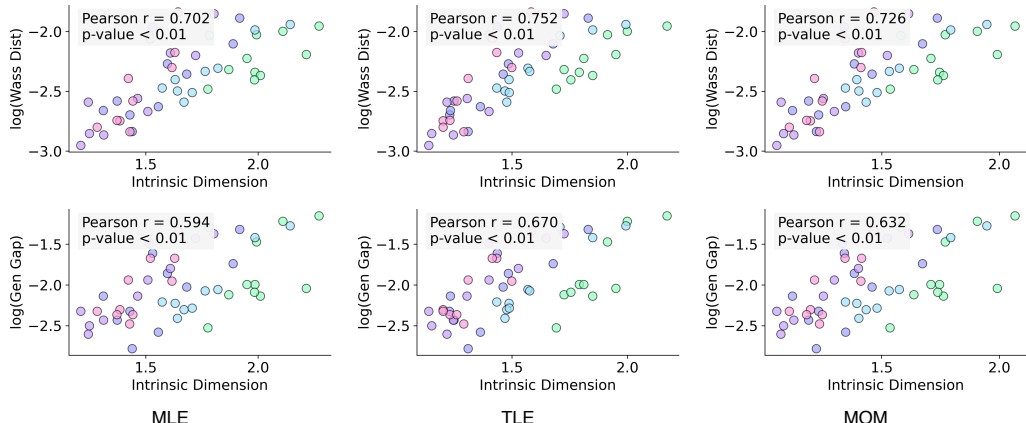

*Figure 8.* **Comparison of dimensionality estimation algorithms on CIFAR-10 embeddings.** Despite using different algorithms, estimated dimensionalities consistently correlate with generalization error, demonstrating robustness to method choice.

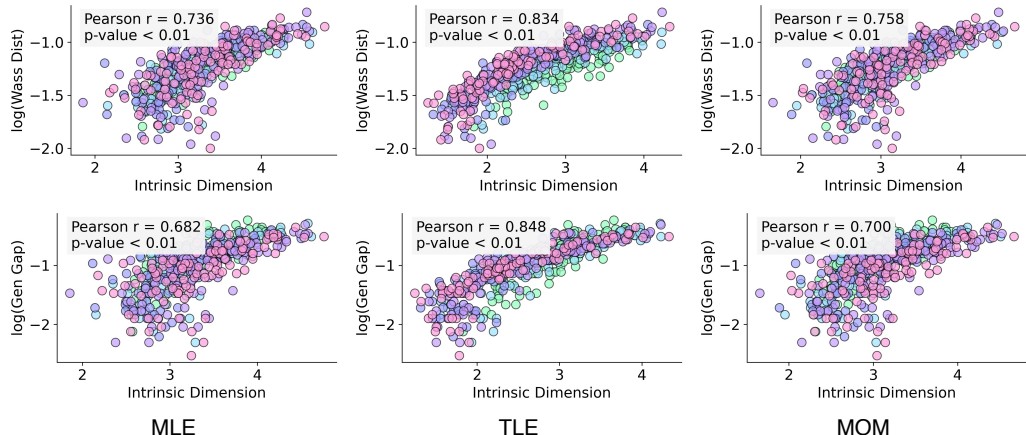

*Figure 9.* **Comparison of dimensionality estimation algorithms on CIFAR-100 embeddings.** Dimensionality estimates remain significantly associated with generalization error across different algorithms.

## D. Details of Large-Scale Pretrained Models

We provide details on the large-scale pretrained models used to evaluate embedding geometry, covering both vision and language modalities.

**Model families and pretraining regimes.**    We consider two modalities:

*Table 2.* Top-1 accuracy of large-scale pretrained ConvNeXt models on ImageNet-1K. Models span different architectural scales and pretraining regimes, including ImageNet-12K, ImageNet-22K, and CLIP-style LAION pretraining.

| Model | Top-1 Accuracy (%) |
|---|---|
| ConvNeXt-Nano (IN-12K → IN-1K) | 82.88 |
| ConvNeXt-Tiny (IN-12K → IN-1K) | 84.19 |
| ConvNeXt-Tiny (IN-22K → IN-1K) | 84.10 |
| ConvNeXt-Small (IN-12K → IN-1K) | 85.32 |
| ConvNeXt-Small (IN-22K → IN-1K) | 85.75 |
| ConvNeXt-Base (CLIP LAION → IN-1K) | 86.18 |
| ConvNeXt-Large-MLP (CLIP LAION → IN-1K) | 87.34 |
| ConvNeXt-Large (IN-22K → IN-1K) | 87.46 |
| ConvNeXt-XLarge (IN-22K → IN-1K) | 87.37 |
| ConvNeXt-XXLarge (CLIP LAION Soup → IN-1K) | 88.62 |

This collection allows us to probe the geometric–generalization relationship across modalities, architectures, and pretraining scales.

**Methodological considerations.**

- Embeddings from large pretrained models are not strictly i.i.d. due to prior training on massive datasets.
- To approximate the Wasserstein discrepancy between empirical and population embedding distributions, we split the data into two disjoint subsets: one treated as the empirical sample, the other as a proxy for the population distribution.
- Training-set performance is not accessible, so generalization gaps are not directly computed; test accuracy is used as a proxy for downstream performance.

**Metrics.**    Wasserstein distances are computed between disjoint subsets of embeddings, separately for vision and language models, to estimate finite-sample approximation error under a fixed pretrained model.

*Table 3.* Accuracy of pretrained language models fine-tuned on the MNLI benchmark. Models cover a wide range of architectures and parameter scales, including BERT, DeBERTa, DistilBERT, DistilBART, and ALBERT variants.

| Model | MNLI Accuracy (%) |
|---|---|
| BERT-Tiny | 60.00 |
| DistilBERT-Base | 82.00 |
| SciBERT | 83.45 |
| BERT-Base | 84.20 |
| ALBERT-Base-v2 | 85.01 |
| BERT-Large | 86.40 |
| DistilBART (12-1) | 87.08 |
| DistilBART (12-3) | 88.10 |
| DistilBART (12-6) | 89.19 |
| DistilBART (12-9) | 89.56 |
| DeBERTa-Large | 91.30 |
| DeBERTa-XLarge | 91.50 |

## E. Layer-wise Correlations among Dimension, Wasserstein Distance and Generalization Performance

We analyzed embeddings from ResNet-152 at layers 4, 18, 30, 43, 55, 67, 79, 91, 103, 115, 127, 139, and 152, and computed the correlation between embedding dimensionality, Wasserstein distances on the validation and test sets, and generalization error.

Correlations are relatively weak in early layers but increase in deeper layers, with a pronounced rise after layer 140. This suggests that deeper embeddings more faithfully capture features relevant to generalization.

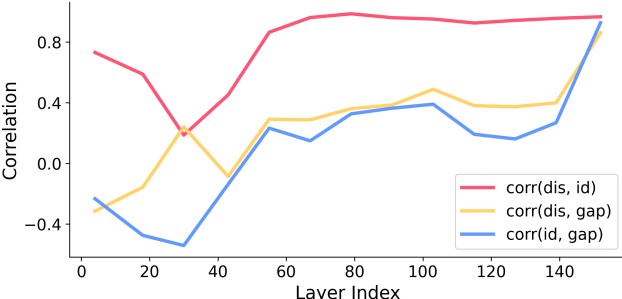

*Figure 10.* **Layer-wise correlations between embedding dimensionality, Wasserstein distance, and generalization error in ResNet-152.** Deeper layers exhibit stronger correlations, indicating the increasing alignment between representation properties and generalization.

## F. Predictive Power of Intrinsic Dimension and Wasserstein Distance

To further examine whether representation-level quantities are informative for generalization beyond the models used to estimate them, we evaluate the predictive power of intrinsic dimension and Wasserstein distance on unseen architectures. Specifically, we use ResNet18, ResNet34, and ResNet50 as source models, and test whether the trends captured by each quantity can predict the generalization behavior of larger unseen models, ResNet101 and ResNet152.

We consider intrinsic dimension and Wasserstein distance separately as post-hoc predictors of the observed generalization gap. This setting evaluates whether each representation-level quantity provides transferable information across model scales, rather than merely reflecting architecture-specific behavior. As shown in Figure 11, both intrinsic dimension and Wasserstein distance exhibit meaningful predictive trends on the held-out architectures. The predicted values broadly follow the measured generalization gaps on both CIFAR-10 and CIFAR-100, suggesting that these quantities capture generalization-relevant properties of the learned embedding distributions.

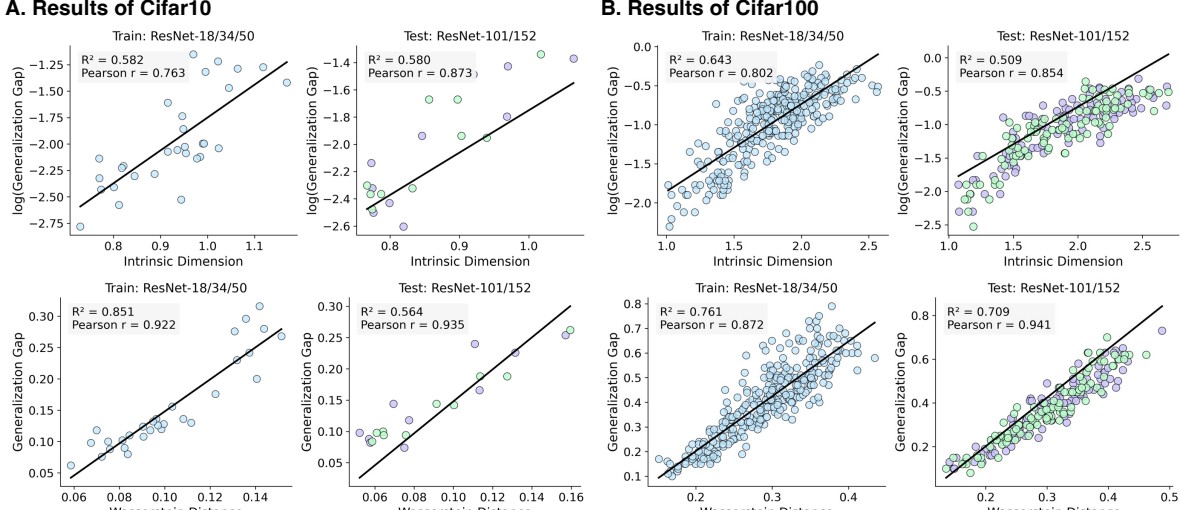

*Figure 11.* **Predicting Generalization Error on Unseen Architectures.** ResNet18, ResNet34, and ResNet50 are used as source models, while ResNet101 and ResNet152 are used as unseen target models. Intrinsic dimension and Wasserstein distance are evaluated separately as predictors of the observed generalization gap. **(A)** CIFAR-10. **(B)** CIFAR-100. Both quantities show meaningful predictive trends across model scales.

These results provide additional empirical support for our theoretical analysis. Intrinsic dimension reflects the effective complexity of the embedding distribution, while Wasserstein distance directly measures the discrepancy between empirical embedding distributions. Their predictive behavior on unseen architectures indicates that representation geometry can serve as a useful post-hoc signal for assessing generalization.

