# OpenReview forum: "Understanding Generalization from Embedding Dimension and Distributional Convergence"
_ICML.cc/2026/Conference — ICML 2026 regular_

### Official Review · Reviewer_7s23 · 2026-02-23

**Soundness:** 3
**Presentation:** 3
**Significance:** 3
**Originality:** 4
**Overall Recommendation:** 4
**Confidence:** 3

**Summary:**

This paper studies deep learning generalization from a representation perspective, bypassing traditional parameter-based capacity metrics. The authors propose bounds on population risk derived from two primary factors: (i) the intrinsic/Wasserstein dimension of embedding distributions (which controls the convergence of empirical to population embedding distributions in Wasserstein distance), and (ii) the Lipschitz sensitivity of downstream mappings. A central claim is that at the final embedding layer, architectural sensitivity vanishes, and generalization is dominated by the embedding dimension. The main theorem provides a dimension-dependent post-training bound with explicit Wasserstein convergence terms, demonstrating convergence at  a certain rate.

**Compliance With Llm Reviewing Policy:**

Affirmed.

**Key Questions For Authors:**

The key question is about a comparison with recent generelaization bounds in particular rank and norm-based bounds (see above). My evaluation assumes a satisfactory answer from the authors. It will become worse if the answers are not satisfactory.


Other questions are:
1) How do you reliably estimate the Wasserstein/intrinsic dimension, and how sensitive is the empirical validation to the choice of estimator and its hyperparameters?
2) Are there counterexamples where the embedding dimension is low, but out-of-sample generalization is still poor? If so, what does your theory predict in those scenarios?
3) Does the bound hold, or does it become entirely vacuous, under scenarios of severe label noise?
4) Can you provide a true "prediction" experiment?
5) How does this framework theoretically and empirically relate to Rademacher complexity bounds? In general, to existing margin-based or compression-based or rank-based or norm-based measures of generalization?

**Limitations:**

The authors must add an explicit discussion regarding the bound constants and the potential for vacuous bounds. They should also extend the discussion on the computational cost of estimating Wasserstein-related quantities at scale are inadequately discussed. Furthermore, the paper should clarify the specific, practical conditions under which the claim that "final layer sensitivity vanishes" actually holds true.

**Strengths And Weaknesses:**

Strengths are:
1) Theory: The theoretical proofs leveraging the upper Wasserstein dimension look rigorous to me. Combining Wasserstein distributional convergence of embeddings with Lipschitz tails is an elegant synthesis.
2) Conceptual Shift: Moving generalization bounds away from vacuous parameter counts and toward representation geometry may well be  a refreshing direction for deep learning theory.
3) Presentation: The distinction between architectural sensitivity at intermediate layers and its disappearance at the final layer is well explained. The overall framing of a "post-training, fixed predictor" provides a clean, comprehensible narrative.
4) Experiments: Validating that empirical embedding distributions actually converge according to the predicted rate provides an empirical foundation for the theoretical claims.

Weaknesses are:
1) Vacuous bounds: As with many generalization bounds, there is a significant risk of vacuous bounds (the experiments are compatible with vacuous bounds). Constants (such as Lipschitz constants and diameters) can easily dominate the bound. The paper needs to discuss this issue and compare with rank-based and norm-based bounds for deep networks. In particular see https://arxiv.org/pdf/2510.21945 and https://arxiv.org/pdf/2301.12033. The latter paper shows non-vacuous norm-based bounds for MNIST.
2) Estimation: Measuring intrinsic/Wasserstein dimension accurately in high-dimensional spaces is notoriously unstable and difficult. The paper needs a more careful explanation of how this is estimated robustly across different datasets and models.
3) Literature Discussion The impact of the paper depends heavily on demonstrating that this is not simply a restatement of known phenomena. It needs stronger arguments against existing representation-based generalization bounds especially the norm and rank ones (see above).

---

> ### Author Rebuttal · Authors · 2026-03-29
>
> We thank Reviewer for the thoughtful feedback. Additional experimental results: https://anonymous.4open.science/r/2026-ICML-CC59/README.md
>
> **1. Vacuity of the bound.**
>
> We agree the bound may be numerically loose. Nevertheless, we believe the result is still meaningful for three reasons.
>
> * First, the constants have clear geometric meaning: Lipschitz constants reflect network sensitivity, and the diameter bounds the spatial extent of representations, none are opaque proof artifacts.
>
> * Second, some constants are controllable through architecture design: Lipschitz constant can be controlled by spectral normalization (Miyato et al., 2018), and diameter can also be restricted through bounded activations (e.g., sigmoid).
>
> * Third, our goal is not to provide a tighter certificate than existing bounds, but to explain why low-dimensional embeddings generalize better. Empirically, we find that dimension and Wasserstein distance consistently track generalization across models, including large-scale settings, suggesting that the bound captures the dominant mechanisms even if the constants may be loose.
>
> **2. Comparison with existing bounds.**
>
> Most existing bounds are parameter-centric. Approaches based on Rademacher complexity, norms of the weight (norm-based bounds, e.g., Ledent et al., 2025; Galanti et al., 2023), and related quantities aim to characterize how the complexity of the model is reflected in parameter space. Galanti et al. provide an impressive example of this line of work by obtaining a non-vacuous norm-based bound on MNIST. At the same time, such tightness relies on specific datasets and architectures (fully-connected networks), and these quantities often become large as model scale increases.
>
> Our perspective is different: rather than analyzing parameters, we analyze learned embeddings. The key complexity term in our bound is the dimension of the representation distribution. Unlike many parameter-space measures, this quantity is not directly tied to parameter count and therefore does not automatically grow with model size. Empirically, we find that it remains informative about generalization across scales, including in large language and vision models. To the best of our knowledge, this is the first generalization bound stated explicitly in terms of learned embeddings, and we view it as complementary to parameter-centric approaches.
>
> At the same time, the two views are closely related. For intermediate layers, our bound depends not only on embedding dimension but also on the Lipschitz sensitivity of the downstream map, which in many architectures can be upper bounded by products of spectral norms. Thus, our result complements norm-based bounds by introducing a representation-level term while remaining directly connected to classical complexity measures. We will revise the paper to make this distinction and connection clearer.
>
> **3. Reliability of intrinsic-dimension estimation.**
>
> We agree that intrinsic-dimension estimation can be challenging in high-dimensional ambient spaces. In our case, however, the target is the intrinsic geometry of learned embeddings, which is typically much lower-dimensional than the ambient space.
>
> In Appendix C, we compare multiple estimators, including MLE, TLE, and MOM, and vary the neighborhood parameter k. We have also added an analysis across different validation splits, shown in **Figure 1 of the anonymous link**, which further demonstrates that the main trends are consistent across splits.
>
> For Wasserstein distance, our empirical conclusions rely on stable relative trends across models/classes, not on a single absolute distance value, which makes them less sensitive to small estimation noise.
>
> **4. Low dimension but poor generalization.**
>
> Yes, and this is explicitly predicted by our theory. Low-dimensional embeddings alone do not guarantee good generalization, because for intermediate layers the bound also depends on the Lipschitz sensitivity of the downstream map. Thus, a model can have low embedding dimension but still generalize poorly if the downstream network strongly amplifies small perturbations. This is exactly what we observe in the width-intervention experiment: narrowing the network lowers intrinsic dimension but can increase the downstream Lipschitz constant, so the generalization gap does not improve monotonically.
>
> **5. Label noise and true prediction experiments.**
>
> Under severe label noise, the bound still holds, but it can become loose because the Bayes noise terms grow and may dominate the Wasserstein term. This reflects increased irreducible difficulty rather than a failure of the analysis.
>
> For a true prediction test, we added a held-out model experiment in **Figure 2 in anonymous link**. We train a simple predictor on one set of models (ResNet18/34/50) and evaluate it on unseen models (ResNet101/152). The results show that dimension and Wasserstein distance have meaningful predictive power for generalization on unseen models.

---

> > ### Author Rebuttal · Reviewer_7s23 · 2026-04-01
> >
> > Thanks for the answer. I think that a small revision in the paper to make the distinction and connection wrt existing bounds clearer would make everything clearer!

---

> > > ### Author Response · Authors · 2026-04-07
> > >
> > > Thank you for the helpful suggestion. We agree that making the relationship to existing bounds clearer will strengthen the paper, and in the revision we will explicitly clarify both the connections and the distinctions between our representation-based framework and prior generalization bounds.

---

### Official Review · Reviewer_s2pL · 2026-03-07

**Soundness:** 4
**Presentation:** 4
**Significance:** 2
**Originality:** 3
**Overall Recommendation:** 5
**Confidence:** 3

**Summary:**

This paper studies generalization from a representation point of view, by analyzing the embedding space produced by a trained neural network. The main idea is that generalization can be related to two things: the intrinsic dimension of the learned embeddings, and the sensitivity of the remaining mapping from embeddings to outputs. This idea is rigorously formulated as a generalization bound. The bound suggests that lower intrinsic dimension can benefit generalization, while large Lipschitz constants in the downstream mapping can hurt generalization. The paper also gives a simpler result for the final layer, where the role of architecture-specific sensitivity becomes smaller.

Experimentally, the paper studies this connection on several models and datasets. The results show that lower-dimensional final-layer embeddings are often correlated with better generalization. The paper also shows that reducing representation dimension alone does not always help, because sensitivity can increase at the same time. Overall, the paper tries to give a geometric and representation-based view of why some neural networks generalize better than others.

**Compliance With Llm Reviewing Policy:**

Affirmed.

**Final Justification:**

The paper is clear and technically solid, and I remain positive about it overall. The main idea about separating representation geometry from downstream sensitivity is interesting, and the rebuttal helped clarify this point more concretely. In particular, I now better understand how the proposed layer-wise quantities are intended to explain why lower-dimensional representations do not automatically lead to better generalization.

The rebuttal improved my understanding, but it did not change my main view of the paper's scope. I still think the framework is best interpreted as a post-training analysis of a fixed learned representation, so its practical scope is narrower than the broadest reading of the paper might suggest. At the same time, I think the paper contains a useful idea that others may build on. Overall, I found the paper meaningful and technically careful, with clear limitations but enough value to justify acceptance.

**Key Questions For Authors:**

see weakness.

**Limitations:**

yes

**Strengths And Weaknesses:**

**Strengths**
- The representation-centered view is interesting. It gives a different angle from standard capacity-based analysis.
- The theory is reasonably clear and the main message is easy to understand: low-dimensional embeddings can help, but stability also matters.
- The experiments are connected to the theory in a fairly direct way. This makes the paper easier to trust.
- The final-layer analysis is a nice part of the paper, and it gives a cleaner interpretation of why final-layer geometry may matter.

 **Weaknesses**
- **The main rate is not really an algorithmic convergence rate.**
  A central issue is that the representation map is not fixed in the actual learning problem. In the paper, the embedding at layer $k$ is treated as if it comes from a fixed trained model, and the embedding feature map is denoted by $F_{\leq k}$.  The generalization bound in the main theorem is determined by the property of $F_{\leq k} $. However, the map $F_{\leq k}$ is itself learned from the training set, so it should really be written as $F_{\leq k,S}$ or $F_{\leq k,n}$. Because of this, the displayed rate should not be interpreted as a true learning curve with respect to sample size. When $n$ changes, the embedding map may also change a lot, and the intrinsic dimension $d_k$ may also change. Therefore the post-training generation bound cannot explain the effectiveness of the algorithm.

- **The framework’s distinctive value is confined to a specific scenario**
A network may compress the training data into a low‑dimensional representation even if the true out‑sample embedding distribution is far more complex. Using training embeddings to estimate intrinsic dimension would therefore be misleading. The paper avoids this by using held‑out data, but then a new issue appears: if that held‑out data has labels, why not simply look at validation/test loss? The geometric quantities become an indirect proxy rather than a primary tool. The approach is most useful when one has a large amount of unlabelled held‑out data (so validation loss cannot be computed) and wants a label‑free proxy for generalization. This narrow scope weakens its claim as a general explanation of deep learning generalization.

---

> ### Author Rebuttal · Authors · 2026-03-29
>
> **1. Meaning of the rate**
>
> We thank the reviewer for pointing out this important distinction. We agree that the $n$-dependence in our theorem should not be interpreted as an algorithmic convergence rate or a learning curve with respect to training set size.
>
> More precisely, the representation map is learned from the training set and should therefore be written as $F_{\le k, S}$ , as the reviewer suggests. We will revise the notation accordingly. At the same time, we would not write it as $F_{\le k, n}$, because the $n$ appearing in our bound does not denote the number of training samples used to learn the network. In the theorem, $n$ is the number of i.i.d. samples used to construct the empirical embedding distribution for a fixed trained model, interpreting it as training set size would be inconsistent with the way the i.i.d. assumption is used in the proof.
>
> Viewed this way, the main rate characterizes how efficiently, for a fixed trained network, the learned embedding captures the underlying data distribution. In particular, a lower intrinsic dimension implies that the embedding-space distribution can be characterized more efficiently from finite samples, yielding faster convergence in Wasserstein distance.
>
> We will revise the paper to make this distinction explicit and avoid any wording that could suggest an algorithmic learning-curve interpretation.
>
> **2. Scope, held-out embeddings, and why not just use validation loss?**
>
> We appreciate this point and agree that using training embeddings to estimate these quantities would be problematic.
> Our analysis is explicitly post-training: the empirical embedding distribution should be formed from samples that are independent of the training set used to learn $F_{\le k, S}$​. This is why we use held-out embeddings rather than training embeddings.
>
> We also agree that when labels are available, validation/test loss is the most direct quantity for model selection. Our aim is not to replace validation loss, but to provide a complementary representation-level diagnostic. In particular, the proposed quantities support layer-wise analysis and help disentangle the roles of embedding geometry and downstream sensitivity, which are not revealed by validation loss alone.
>
> More broadly, we agree with the reviewer that this framework is not a universal replacement for standard validation metrics, nor a complete explanation of deep learning generalization. Its contribution is better viewed as a post-training, representation-centric diagnostic tool, complementing rather than replacing standard validation metrics, and is especially valuable when label-free (e.g., comparing self-supervised or pre-trained checkpoints before downstream fine-tuning) or layer-wise insights are needed beyond aggregate prediction loss.

---

> > ### Author Rebuttal · Reviewer_s2pL · 2026-04-01
> >
> > Thanks for the clarification. I only have one following question:
> >
> > >In particular, the proposed quantities support layer-wise analysis and help disentangle the roles of embedding geometry and downstream sensitivity, which are not revealed by validation loss alone.
> >
> > Could you offer some concrete examples (e.g. what kinds of downstream sensitivity, how to utilize your method in this set up)?
> >
> > ----
> >
> > Thank you for the clarification. I now better understand the intended role of the layer-wise quantities in separating embedding geometry from downstream sensitivity.
> >
> > I would like to suggest a more concrete application setting that seems especially well matched to the paper: **pretrained VAE or RAE [1]/ latent diffusion models for image generation**.
> >
> > In my opinion, this setting fits the paper better than the current classification-style examples. The key reason is very concrete. In a latent generative model, it is not enough for the latent representation to be low-dimensional or well compressed. The model must also generate **new images**, not just reconstruct points near the training set. This makes downstream sensitivity particularly important. If the decoder is too sensitive, then a latent code that moves even slightly away from the support of the pretrained latent distribution can produce visibly poor images, artifacts, or unstable semantics. This is exactly the kind of tradeoff your framework is trying to capture: a latent representation may look favorable from the viewpoint of compression or intrinsic dimension, while the downstream decoder may amplify small off-support perturbations in a harmful way.
> >
> > This issue seems especially important in latent diffusion. The diffusion model operates in latent space and necessarily explores many latent points that are not exactly training examples. So a good latent space should not only be compact; it should also support a decoder whose behavior remains stable when the latent variable moves to nearby but genuinely new regions. In this sense, the paper's message is very relevant here: **low-dimensional representation alone is not enough; one also needs controlled downstream sensitivity**.
> >
> > For this reason, I think an analysis of pretrained VAE latents under your framework could be very meaningful. This would make the practical meaning of the framework much more concrete. In fact, I think such an analysis may have more impact than some of the current ResNet-style experiments, because image generation is a setting where sensitivity of the downstream map is not a side issue, but a central one.
> >
> > If a careful preliminary analysis in this direction is feasible during the rebuttal period, it would significantly strengthen the paper in my view. If this is not feasible within the current timeline, I would still strongly encourage the authors to discuss this setting explicitly in the paper, since it seems like a particularly natural and important use case for the proposed framework.
> >
> > At present I still view the paper positively overall and keep my current recommendation, but I wanted to emphasize this suggestion because I think it could increase both the practical relevance and the conceptual sharpness of the work.
> >
> > ---
> > [1] https://openreview.net/forum?id=0u1LigJaab

---

> > > ### Author Response · Authors · 2026-04-01
> > >
> > > Thanks for the follow-up.
> > >
> > > By downstream sensitivity, we mean the sensitivity of the trained tail mapping from a layer-k embedding to the final prediction/loss. Concretely, it quantifies how much a small perturbation in the embedding is amplified by the remaining layers. In practice, this can be locally characterized by the Jacobian operator norm of the tail network, and upper bounded by spectral-norm-based quantities.
> > >
> > > A concrete example is the ***narrow-but-deep setting in Section 5.4***. A common intuition is that forcing a lower-dimensional intermediate representation should improve generalization. Our experiments show why this can fail: narrowing a layer does reduce the intrinsic dimension of its embedding, but it can simultaneously increase the sensitivity of the downstream mapping, since the later layers often need to amplify a more compressed signal to preserve task-relevant information. These two effects can offset each other, so the performance need not improve monotonically.
> > >
> > > This also illustrates how the method is intended to be used. When comparing fixed architecture variants or trained checkpoints, one can examine whether a change improves representation geometry (e.g., lower intrinsic dimension or smaller empirical distributional discrepancy) at the cost of a more sensitive tail network. Validation loss only reports the net outcome, whereas our framework helps separate these contributions in a layer-wise way. We will revise the text to make this interpretation more explicit.

---

### Official Review · Reviewer_fkij · 2026-03-12

**Soundness:** 4
**Presentation:** 4
**Significance:** 3
**Originality:** 3
**Overall Recommendation:** 5
**Confidence:** 3

**Summary:**

This paper introduces a representation-centric framework to explain how highly over-parameterized neural networks generalize. The authors demonstrate that a model's generalization error is jointly controlled by the intrinsic dimension of its learned embeddings and the Lipschitz sensitivity of its downstream layers. Their approach explains why the geometric complexity of a network's final layer acts as a strong empirical predictor of its overall performance.

**Compliance With Llm Reviewing Policy:**

Affirmed.

**Final Justification:**

The authors effectively addressed my concerns, and the rebuttal helped me better understand their contribution. I maintain my positive assessment.

**Key Questions For Authors:**

1. How restrictive is assumption 3.13 and in particular could you provide examples of excluded settings?
2. The experiments demonstrate a strong linear correlation between the final-layer intrinsic dimension and the generalization gap. Since the theory establishes an upper bound with admittedly loose constants, does this empirical result suggest the bound is tighter in practice than the mathematics imply?

**Limitations:**

Yes

**Strengths And Weaknesses:**

### Strengths

The paper is technically sound, clear and well-structured. The results offer useful insights on the interplay between generalization and intrinsic embedding dimension, and are backed by robust empirical evidence, successfully validating the claims across controlled architectural interventions as well as large-scale vision and language models. Crucially, their framework does not require labeled data, and could be applied to unsupervised learning and pretraining problems.

### Weaknesses

The notable weaknesses are primarily practical and theoretical limitations, most of which are already acknowledged by the authors, that do not severely detract from the main contributions. In particular
1. The authors explicitly note that their generalization bound contains constants that may be loose.
2. Computing exact Lipschitz constants for deep networks is intractable in general. The experiments are forced to rely on proxies that overestimates such constants. This issue can penalize the practicality of the results. (See Questions)
3. The theoretical proofs rely on the assumption that the Bayes predictor is locally Lipschitz continuous, but the restrictiveness of this assumption is not sufficiently discussed or explored.

---

> ### Author Rebuttal · Authors · 2026-03-29
>
> We thank the reviewer for the thoughtful comments and constructive feedback.
>
> **1.Are the constants too loose?**
>
> We thank the reviewer for this question. We agree that the constants in the bound may be loose. This is not unusual in deep learning: Jiang et al. show that many generalization bounds are numerically vacuous in practical regimes, even though the corresponding complexity measures can still strongly correlate with generalization [1].
>
> In our case, the constants are at least interpretable rather than arbitrary; they involve the Lipschitz constants of the network and Bayes predictor, the Wasserstein convergence constant  $C_k$, and the embedding diameter $D_k$. Some of these quantities are also practically controllable. For example, in fully connected ReLU networks, the Lipschitz constant can be upper bounded by the product of spectral norms of the weight matrices.
>
> Most importantly, our claim is not that the bound is numerically tight, but that its structure identifies the key factors governing generalization. The constants do not affect the main scaling $n^{-1/(d_k + \epsilon)}$, and empirically we observe that intrinsic dimension and Wasserstein distance consistently track generalization across models. We therefore view the result as informative at the level of identifying the right controlling quantities, even if the absolute bound may be loose. We will clarify this point in the revision.
>
>
> **2.Does the need to estimate Lipschitz constants via proxies limit practicality?**
>
> We agree that exact computation of Lipschitz constants for deep networks is generally intractable, and practical proxies often provide conservative upper bounds. This is a known limitation and remains an important open problem.
>
> However, in our framework, the Lipschitz term plays primarily a mechanistic role: it quantifies how perturbations in the learned embedding are amplified by the downstream mapping. Because exact evaluation is generally infeasible for deep networks, the proxies used in our experiments should be viewed mainly as exploratory tools rather than precise estimates.
>
> Moreover, in Corollary 4.2, we show that in the final-layer setting the downstream map reduces to the identity, ***eliminating the need for Lipschitz constant estimation***. This yields a practically useful regime in which generalization is governed mainly by embedding geometry and distributional convergence.
>
>
> **3.How restrictive is the local Lipschitz assumption on the Bayes predictor? What does it exclude?**
>
> Assumption 3.13 consists of two parts with different levels of restrictiveness.
>
> For the network map $F_k$, local Lipschitz continuity is mild and holds for standard architectures (e.g., compositions of linear layers with ReLU) on bounded domains. The more substantive requirement is the local Lipschitz continuity of the Bayes predictor $F_k^*$, which assumes that the conditional label distribution varies smoothly with the representation in the region supported by the data.
>
> This assumption is local rather than global: it only needs to hold in a neighborhood of the embedding support. Intuitively, it excludes settings where small changes in the embedding induce large, discontinuous changes in the label posterior.
>
> Examples of excluded cases include:
>
> (i) sharp or near-step discontinuities in class posteriors,
>
> (ii) strongly discontinuous or feature-dependent label noise, and
>
> (iii) degenerate representations where nearly identical embeddings correspond to very different label distributions.
>
> We will clarify this discussion in the revision.
>
> **4.Does the strong linear correlation at the final layer imply that the bound is tighter in practice than the mathematics suggest?**
>
> We thank the reviewer for this insightful question. We do not interpret the empirical correlation as evidence that the full upper bound is numerically tight term by term. Instead, we view it as evidence that the main mechanism identified by the theory is active in practice, namely that embedding dimension influences Wasserstein convergence, which in turn affects generalization.
>
> This is especially natural in the final-layer setting. The theory shows that intermediate layers involve both embedding geometry and downstream sensitivity, while at the final layer the downstream mapping disappears and the dependence on embedding geometry becomes more direct. A strong correlation between intrinsic dimension and generalization is therefore consistent with the theoretical prediction.
>
> Importantly, such empirical trends reflect the leading-order dependence captured by the bound, even if constant factors are loose. In this sense, the result should be understood as supporting the relevance of the structure identified by the theory, rather than indicating that the bound is tight in a numerical sense.
>
> **Reference**
>
> [1] Jiang, Yiding, et al. "Fantastic Generalization Measures and Where to Find Them." International Conference on Learning Representations.

---

> > ### Author Rebuttal · Reviewer_fkij · 2026-04-01
> >
> > Thanks for your replies and clarification. I confirm the current score.

---

> > > ### Author Response · Authors · 2026-04-07
> > >
> > > Thank you for your thoughtful follow-up and for confirming the current score. We sincerely appreciate your careful reading of our responses and your constructive feedback throughout the review process. Your comments have been very helpful in improving the clarity and presentation of the paper.

---

### Decision · Program_Chairs · 2026-04-30

**Decision:**

Accept (regular)

**Comment:**

The paper develops a representation-centric perspective on generalization by relating population risk to two factors: the intrinsic (Wasserstein) dimension of learned embeddings and the Lipschitz sensitivity of the downstream mapping. Reviewers found the paper technically solid, clearly written, and conceptually interesting, with a useful distinction between embedding geometry and downstream sensitivity, especially in the final-layer setting. The rebuttal addressed several concerns well.

My main reservation is about the practical strength and predictive power of the bound itself. In particular, the O(n^{-1/(d_k+\varepsilon)}) rate is slower than typical norm-based or parameter-based bounds and still suffers from the curse of dimensionality. While this type of rate is standard for Wasserstein convergence and is therefore not a flaw specific to this paper, it does limit the sharpness of the resulting guarantee as a quantitative certificate. Relatedly, the framework seems most compelling as an explanatory lens, rather than as a tight predictive bound on generalization. That said, the reviewers were broadly positive, and the rebuttal clarified the intended scope and novelty of the contribution. Overall, I find the paper meaningful and technically careful to recommend acceptance, provided the final version clearly incorporate suggestions from the reviewers.